# Metabolomic Profiling Identifies Key Metabolites and Defense Pathways in *Rlm1*-Mediated Blackleg Resistance in Canola

**DOI:** 10.3390/ijms26125627

**Published:** 2025-06-12

**Authors:** Xiaohan Zhu, Peng Gao, Shuang Zhao, Xian Luo, Liang Li, Gary Peng

**Affiliations:** 1Saskatoon Research and Development Center, Agriculture and Agri-Food Canada, 107 Science Place, Saskatoon, SK S7N 0X2, Canada; peng.gao@agr.gc.ca; 2The Metabolomics Innovation Centre and Department of Chemistry, University of Alberta, Edmonton, AB T6G 2G2, Canada; szhao1@ualberta.ca (S.Z.); xluo2@ualberta.ca (X.L.); liang.li@ualberta.ca (L.L.)

**Keywords:** blackleg, canola, metabolomics, resistance, *Rlm1*, Topas

## Abstract

Blackleg disease poses a major threat to global canola production. The resistance gene *Rlm1*, corresponding to the avirulence gene *AvrLm1* in the pathogen *Leptosphaeria maculans*, has been widely used to mitigate the impact of the disease. To investigate the biochemical basis of *Rlm1*-mediated resistance against blackleg, we conducted an LC-MS–based analysis of a susceptible Topas double haploid (DH) line and its isogenic *Rlm1*-carrying resistant counterpart for metabolomic profiles during the infection process. Samples were labeled with ^12^C- and ^13^C for LC-MS analyses to enhance both chemical and physical properties of metabolites for improved quantification and detection sensitivity. Resistant plants showed early and sustained accumulation of several defense metabolites, notably pipecolic acid (PA, up to 326-fold), salicylic acid (SA), and gentisic acid (GA) in *L. maculans*-inoculated Topas–*Rlm1* plants compared to mock-inoculated Topas–*Rlm1* controls (adjusted *p* < 0.05), indicating activation of lysine degradation and hormonal defense pathways. Elevated glucosinolates (GLS), γ-aminobutyric acid (GABA), and melatonin precursors may further contribute to antimicrobial defense and cell-wall reinforcement. In contrast, flavonoid and phenylpropanoid pathways were down-regulated, suggesting metabolic reallocation during resistance. Exogenous application of PA, SA, GA, ferulic acid, and piperonylic acid (a known inhibitor of the phenylpropanoid pathway in plants) significantly reduced infection in susceptible canola varieties, validating their defense roles against blackleg. These results offer new insights into *Rlm1*-mediated resistance and support metabolic targets for breeding durable blackleg resistance in canola.

## 1. Introduction

Canola/rapeseed (*Brassica napus* L.) is a major cash crop in Canada, with national production reaching 17.8 million tons in 2024 and contributing nearly CAD 40 billion to the Canadian economy [1]. Blackleg, caused by *L. maculans* Ces. and de Not., poses a serious threat to canola production in Canada, with estimated global economic losses exceeding USD 900 million annually [2,3]. Effective management of blackleg is therefore critical for a successful canola crop.

Genetic resistance, in combination with extended crop rotation, is the key strategy for blackleg management in Canada, where many canola cultivars carry the specific resistance (*R*) genes *Rlm1*/*LepR3* and *Rlm3* [4], while almost all cultivars also possess a level of quantitative resistance [5,6]. The R gene *Rlm1* has been deployed in *B. napus* cultivars since the mid 1990s [7]. As more *R* genes are being deployed in new cultivars that target prevalent avirulence (*Avr*) genes in the pathogen population [8,9,10], resistance resilience ought to be monitored because of frequent sexual recombination of *L. maculans* and the high evolutionary potential of the pathogen [11,12]. For instance, *Rlm1* was broken only a few years after its release in France, with a significant decline in the corresponding *AvrLm1* from 83% to <13% in the pathogen population [12]. In western Canada, *Rlm1* is no longer effective since 2007 due to the low presence of the corresponding avirulence gene *AvrLm1* in the pathogen population [8,9]. To date, 22 specific R genes have been reported for blackleg resistance, of which 5 have been successfully cloned (reviewed by Borhan, Van de Wouw [13]). The effective deployment of *R* genes depends mostly on the *Avr* profile of the *L. maculans* population, but a better understanding of the resistance mode of action associated with specific *R* genes can improve our knowledge of host–pathogen interactions in relation to resistance deployment and resilience.

Advances in next-generation sequencing (NGS) and other “omics” technologies have helped identify preliminary mechanisms underlying disease resistance in canola [14,15]. However, such information has not been available for most of the blackleg R genes, especially for key metabolites and defense pathways involved in ‘gene-for-gene’ interactions. In an earlier report, Fudal, Ross [16] described that *L. maculans* appeared to delete *AvrLm1* to regain the virulence toward *Rlm1* worldwide. In a separate study, transcriptome analysis revealed that *Rlm1*-mediated resistance functions locally rather than systemically and involves activation of both salicylic acid (SA) and jasmonic acid (JA) signaling pathways [17]. These findings offer foundational insights into the mechanisms of *Rlm1* resistance and its potential vulnerability to shifts in pathogen populations. Building on this, metabolomic analysis may uncover key metabolites and defense-related pathways directly linked to *Rlm1* function, providing a deeper understanding of its role in blackleg infection and disease progression.

During co-evolution, many plants have developed resistance to diseases, which can often be triggered by pathogen-associated molecular patterns and effectors [18,19,20]. As part of their defense responses, plants can produce a range of low-molecular-weight compounds [21], many of which are secondary metabolites belonging broadly to flavonoids (Flav) and phenolics, alkaloids and sulfur-containing compounds, and terpenoids [22,23]. Many of them play a role in plant defense against pathogens [24,25]. Metabolomic study of canola–*L*. *maculans* interactions can deepen our knowledge of blackleg resistance, uncover novel plant defense compounds and pathways, and support disease resistance breeding efforts by targeting key metabolites or their associated genes.

Metabolomic studies commonly employ high-performance liquid chromatography (HPLC), ultra-performance liquid chromatography (UPLC), gas chromatography (GC), and nuclear magnetic resonance (NMR), often coupled with mass spectrometry (MS) [26]. Among these, LC–MS is widely used due to its high sensitivity in detecting metabolites [27]. However, LC–MS can have limitations in terms of metabolite coverage and quantitative accuracy [27]. To address these challenges, chemical isotope labeling (CIL) techniques—such as differential ^12^C-/^13^C-isotope dansylation—have been used to help improve the quantification and sensitivity of metabolomic analysis [28]. When combined with MS, CIL enables broader metabolite coverage across various species and tissue types [27,29,30,31].

Several key metabolites identified through metabolomics have been implicated in plant defense responses. For example, amino acids such as lysine (Lys) play a regulatory role in the activity of defense-related proteins, including β-1,3-glucanase, chitinase, and enzymes involved in reactive oxygen species (ROS) regulation [32]. Lys also contributes to systemic acquired resistance (SAR) by serving as a precursor to pipecolic acid (PA) [33], a known SAR signaling molecule that functions alongside salicylic acid (SA) [34] and redox-related mechanisms [35]. Additionally, the accumulation of γ-aminobutyric acid (GABA) has been associated with enhanced disease resistance in rice, Arabidopsis, and other plant species [36,37,38]. Methionine (Met) cycle enzymes, along with ethylene (ET) and polyamines, are also involved in resistance to viral pathogens [39]. Furthermore, gentisic acid (GA) has been shown to activate peroxidase activity in cucumber and purple velvet (*Gynura aurantiaca*) following pathogen infection or SA treatment [40].

Here, we report a study investigating the metabolomic basis of *Rlm1*-mediated blackleg resistance in canola using the CIL LC–MS platform. By identifying key metabolites and associated post-transcriptional defense-related pathways, this research seeks to enhance our understanding of the molecular mechanisms underlying resistance. Building on our previous transcriptome analysis of *Rlm1*-mediated responses [17], our specific objectives were as follows: (i) characterize metabolomic changes during the infection process; (ii) identify metabolic pathways that contribute to host defense; and (iii) validate the functional roles of highly accumulated metabolites involved in *Rlm1*-mediated resistance to blackleg disease in canola.

## 2. Results

### 2.1. Multivariate Analysis of Metabolomic Data

Multivariate analysis using PCA was performed to identify distinct metabolomic clustering patterns between resistant and susceptible lines following infection, reflecting treatment- and time-specific metabolic responses. More than 3000 metabolites were identified in susceptible (Topas) and resistant (Topas–*Rlm1*) canola varieties with or without *L. maculans* (*AvrLm1*) inoculation. For clarity, these treatments were labelled as TC (Topas-control), TI (Topas-inoculated), RC (*Rlm1*-control), and RI (*Rlm1*-inoculated), corresponding to 3, 7, or 11 days post-inoculation (dpi). The first two principal components of PCA (PC1: 28.2%, PC2: 13.9%) explained the variance (Appendix A). Samples from each treatment and time point formed distinct clusters, demonstrating the data reproducibility. Additionally, QC samples clustered together, indicating high experimental consistency.

At 11 dpi, the TI and RI groups were distinct from other groups and were also separated from each other along PC1 (Appendix A). Additionally, the RC-3dpi, RI-3dpi, and RI-7dpi groups were also distinct from the rest. Notably, all RI samples were separated from those of TI at each sampling point. Relatively, PC1 accounted for most of the variance across time points, while PC2 captured the majority of the variance among treatments.

### 2.2. Univariate Analysis of Metabolomic Data

Univariate statistical analysis was conducted to identify metabolites that were differentially accumulated or suppressed across time points and treatments. Between the control groups TC and RC, a total of 591, 586, and 452 metabolites exhibited increases, while 436, 365, and 385 showed decreases at 3, 7, and 11 dpi (Appendix A). However, between inoculated RI and its water control RC, 259, 457, and 687 metabolites increased, while 469, 392, and 692 decreased at the respective time points. For TI and TC, the corresponding numbers were 169, 317, and 934 (increase) and 181, 198, and 644 (decrease).

Compared to TI, the resistant RI responded more rapidly to infection, with 259 and 457 up-regulated DAMs at 3 and 7 dpi (Appendix A). In contrast, TI displayed only 169 and 317 up-regulated DAMs relative to its control TC at these two time points. In the meantime, RI showed 469 and 392 down-regulated DAMs, while TI had only 181 and 198, relative to their respective controls at 3 and 7 dpi. At 11 dpi, however, TI exhibited the highest number of up-regulated DAMs (934) among all treatments.

### 2.3. DAMs in Relation to Inoculation and Resistance

Venn diagram analysis was used to compare up- and down-regulated DAMs across time points and treatments in resistant and susceptible canola lines, with or without *L. maculans* inoculation, at various infection stages. A significant number of DAMs were shared between RI and its control RC across the time points, indicating the involvement of many of the same metabolites in the Topas–*Rlm1*, with or without infection across 3–11 dpi (Figure 1A–D). This pattern, however, was much less pronounced between TI and TC.

Further comparisons among samples collected at the same dpi showed that only a limited number of DAMs were shared between RI and TI at 3 and 7 dpi (Figure 2A–D). However, at 11 dpi, the number of overlapping up- and down-regulated DAMs increased sharply between the two treatments, reaching 450 and 277, respectively (Figure 2E,F), indicating that many DAMs present in RI were also found in TI at the later stage of infection.

### 2.4. Prominent DAMs and Their Related Pathways

To identify pathways associated with different infection stages in the Topas–*Rlm1*, significantly regulated DAMs at each stage were subjected to pathway analysis. Following the initial analyses above, the most significantly regulated DAMs associated with resistance were identified and further analyzed for potential pathways involved. Out of 1145 up- and 1466 down-accumulated relevant peaks, only 299 and 237 had a single confident match to a known compound in the CIL standard (Tier-1) [41], LIL (Tier-2) [42], and MCIDL (Tier-3) [43]. The remaining peaks either lacked matches or had multiple ambiguous matches and were therefore classified as unknown compounds.

Many of the DAMs exhibited earlier responses in RI samples compared to TI, as shown by the heatmaps of Hierarchical Cluster Analysis (Figure 3). Several DAMs displayed significant regulation only between RI and TI. Out of the significant peaks from 3-dpi samples, only 109 matched known metabolites, with 24 being identified by the CIL standard, 47 by LIL, and 38 by MCIDL. To ensure reliability, only DAMs identified using Tier-1 and Tier-2 libraries were analyzed for pathways in the *Arabidopsis* database [44].

At 3 dpi, several DAMs were significantly regulated in the *Rlm1*-mediated resistance (RI) compared to RC (mock). These metabolites are associated with the biosynthesis of Lys (meso-2,6-diaminoheptanedioic acid) and anthocyanin-ACN (pelargonidin 3-O-β-D-sambubioside), degradation of Lys (PA), and metabolism of purine (adenosine), pyrimidine (cytidine), arginine, and proline [N-carbamoylputrescine (NCP)] (Table 1). In contrast, several DAMs were significantly down-regulated in the RI, including those involved in the biosynthesis of flavones and flavonols [kaempferol 3-O-glucoside, quercitrin, kaempferol, quercetin 3-O-rhamnoside 7-O-glucoside (QRG)], Flav [kaempferol, (-)-epicatechin (EC)], ubiquinone and other terpenoid-quinones [tyrosine (Tyr)], isoquinoline alkaloids (Tyr), and indole alkaloids [tryptophan (Trp)] (Table 1). However, several DAMs identified in RI at 3 dpi were not enriched during pathway analysis, including salicylate β-D-glucose ester (bound SA), (-)-medicarpin, pyrazolidine (PZD), E-6′-HF, γ-glutamyl-β-amino-propiononitrile (γ-Glu-β-APN), m-coumaric acid, and trans-2,3-dihydroxy-cinnamate (t-2,3-DHC).

At 7 dpi, 178 of the 849 distinguished peaks matched known metabolites, and the most significantly regulated DAMs in RI are involved in the metabolism of glutathione (GSH, GSSG, γ-glutamylcysteine), Trp [5-hydroxykynurenamine (5-HKA), serotonin], arginine and proline [γ-aminobutyric acid (GABA), 4-hydroxyproline], pyrimidine (cytidine), and cysteine and methionine-Met [cystathionine (Cth), Met], as well as the biosynthesis of tropane (Trop), piperidine (Pid) and pyridine alkaloid (5-aminopentanal). Some are also involved in the biosynthesis of Lys (Diaminopimelic acid -DAP), Tyr, glucosinolates (GSLs), and ACN, as well as the degradation of Lys [aminoadipic acid (AAA), PA, saccharopine (SAP)] (Table 2). Other DAMs showed decreases in RI, especially those involved in the biosynthesis of flavones and flavonols (quercitrin, luteolin), phenylpropanoids-PPs (ferulate), and arginine (Arg), as well as in the metabolism of glycine (Gly), serine (Ser), and threonine (Thr). Additionally, E-6′-HF, salicylic acid (SA), bound SA, GA, Nε,Nε-dimethyllysine, 5-aminopentanoic acid (5-APA), 3-hydroxymandelic acid (3-HMA), phloroglucinol (PG), dityrosine (DiY), 2,5-dihydroxypyridine (2,5-DHP), emodin (EMD), malonylgenistin (MG), 3,6,7,4′-tetramethylquercetagetin (TMQ), aminoacetaldehyde (AALD), and γ-L-glutamylputrescine (GGP) also increased, while 5-aminolevulinic acid, prolyl-Gly, 3,4-dihydroxybenzaldehyde (3,4-DHBA), and sodium dehydroacetic acid decreased in the RI-7dpi.

At 11 dpi, 287 out of 1379 distinguished peaks from the RI matched known metabolites and the most up-regulated DAMs involved in the biosynthesis of Trop, Pid, and 5-aminopentanal; Lys (DAP), GSLs [homomethionine (homo-Met), p-hydroxyphenylacetothiohydroximate (p-HPAH), phenylalanine (Phe)], ACN (cyanidin 3-O-β-D-sambubioside), Phe, Tyr, and Trp [Phe, 3-(4-Hydroxyphenyl)pyruvate (3-HPP), 2-aminobenzoic acid (2-AA)] also showed significant accumulation (Table 3). Also, DAMs associated with the metabolism of Trp [N-acetylserotonin (NAS), 5-HKA, serotonin], Tyr [3,4-dihydroxymandelaldehyde, homogentisate (HGA), 3-HPP], Phe (Phe) and purines (adenosine monophosphate, hypoxanthine), and Arg and proline (GABA, NCP), as well as Lys degradation (AAA, PA, SAP), were up-regulated relative to the RC control (Table 3). In contrast, DAMs related to biosynthesis of PPs [5-O-caffeoylshikimic acid (5-O-CFSA), ferulate, caffeate], flavones/flavonols (luteolin, QRG), and Flav (p-coumaroyl quinic acid (p-CQA), EC) were down-regulated in RI.

Several DAMs not enriched in the Arabidopsis database but significantly up-regulated in RI included GA, bound SA, SA, PG, methylcysteine, 2-(methylamino)BA, DiY, PZD, AALD, 3-formylsalicylic acid, 3-dechloroethylifosfamide, mangiferin, GGP, MG, E-6′-HF, 2,5-DHP, S-ribosyl-L-homocysteine, TMQ, and dimethylamine. Conversely, kaempferol 3-O-(6″-O-p-coumaroyl)-glucoside, trans-2,3-dihydroxycinnamate (trans-2,3-DHC), bisdemethoxycurcumin (BDMC), and chlorogenic acid (CGA) were the most significantly down-regulated.

### 2.5. Metabolites/Pathways Potentially Related to Rlm1-Mediated Resistance

To clarify the metabolic responses underlying *Rlm1*-mediated resistance, key functional groups were identified based on pathway enrichment and patterns of metabolite accumulation using the *Arabidopsis* database [44] or previously reported data [42,43], covering all time points. In RI, biosynthesis pathways for Lys, GSL, GABA, bound SA, SA, GA, melatonin (Mel), ACN, scopoletin (SCF)/isoscopoletin (IsoScp), and metabolism pathway of NaN, Trp, Tau/HTau, Phe, and amino acids were most activated between 3 and 11 dpi, compared to non-inoculated resistant (RC) or inoculated susceptible (TI) (Figure 4).

#### 2.5.1. Lysine Metabolism and Degradation

Between 3 and 11 dpi, twelve metabolites associated with Lys biosynthesis or degradation were significantly up-regulated in RI (Figure 4A,B). These included PA, SAP, AAA, Nε,Nε-dimethyllysine, N6-acetyl-lysine, 5-APA, 5-hydroxylysine, DAP, and meso-2,6-DAP. Notably, PA exhibited a dramatic increase in RI compared to RC, with fold changes ranging from 164 to 326 across the time points, and was also higher in RI than in TI (Figure 4A). Additionally, peak 1861, tentatively identified as D-1-piperideine-2-carboxylic acid and (S)-2,3,4,5-tetrahydro-piperidine-2-carboxylate (based on Tier-3 library matches), also increased substantially by 11 dpi (Figure 4A). Both compounds are likely intermediates in Lys degradation pathways [61,62].

#### 2.5.2. Defense Signaling Molecules

Metabolites involved in defense signaling, including GA, GABA, SA, and bound SA, were among the most highly accumulated compounds in RI during the infection period (3–11 dpi), with GA reaching peak levels at 11 dpi (Figure 4H,J). The accumulation of benzoic acid (BA) was comparable in RI and TI, but both had lower BA levels than RC, which exhibited the highest levels among all treatments (Figure 4J & Appendix A).

#### 2.5.3. Antimicrobial Metabolites

Several DAMs known for antimicrobial activity were strongly up-regulated in RI (Figure 4K). These included PG, E-6′-HF, MG, 2-aminophenol (2-AP), 2-AA, EMD, PZD, 5-HKA, NCP, and γ-Glu-β-APN. E-6′-HF exhibited a striking 27- to 580-fold increase in RI relative to RC between 3 and 11 dpi (Figure 4E,K). PG, 2-AA, 2-AP, γ-Glu-β-APN, mangiferin, and (-)-medicarpin also showed significant increases in RI, but not in TI, when compared to their respective controls (Figure 4K).

#### 2.5.4. Amino Acid and Secondary Metabolite Pathways

Several DAMs associated with amino acid and secondary metabolite pathways were elevated in RI (Figure 4G,N,O,Q). These included metabolites involved in Trp, NaN, Tau/HTau, and Phe metabolism. In Trp metabolism, NAS, 5-HKA, and 5-hydroxyindoleacetylglycine were significantly increased in RI (Figure 4N). Peak 3324, identified as either IAA or 5-HIAL, is a potential intermediate in Trp/IAA biosynthesis [63,64] and showed increased levels in RI. Additional increases were observed for nicotinic acid mononucleotide and 2,5-dihydroxypyridine (NaN metabolism), Tau, HTau, and aminoacetaldehyde (AALD; Tau/HTau metabolism), as well as for Phe and SA (Figure 4G,O,Q).

Several biosynthetic pathways were also activated in RI, including ACN (Figure 4C), GSL (Figure 4D), Mel (Figure 4F), Trop/Pid/pyridine (Figure 4L), and Phe/Tyr/Trp biosynthesis (Figure 4M). For example, five DAMs related to ACN biosynthesis, including pelargonidin 3-O-β-D-sambubioside, accumulated in RI at multiple time points (Figure 4C). GSL-related metabolites such as homo-Met, p-HPAH, Phe, Met, and Leu also increased in RI (Figure 4D).

In Mel biosynthesis, NAS and 5-MT were significantly increased at 11 dpi in RI (Figure 4F). Additionally, peak 3618—putatively matched to Mel in the Tier-3 library—increased by 130- and 389-fold in RI compared to RT at corresponding time points (Figure 4F). Other compounds, including 5-aminopentanal (Trop/Pid/pyridine biosynthesis), D-erythrose 4-phosphate, and 3-HPP (Phe/Tyr/Trp biosynthesis), were also strongly up-regulated in RI (Figure 4L,M).

#### 2.5.5. Scopoletin/Isoscopoletin Biosynthesis

The highly induced metabolite E-6′-HF, along with ferulic acid (FA) and caffeic acid (CFA), are key intermediates in the scopoletin biosynthesis pathway and may also be involved in isoscopoletin production [65,66]. In canola, FA and CFA accumulated more strongly in RC than in other treatments (Figure 4E & Appendix A), whereas E-6′-HF showed a consistent and marked increase in RI at all time points (Figure 4E).

#### 2.5.6. Redox Metabolism (GSH/GSSG)

Both reduced (GSH) and oxidized glutathione (GSSG) levels increased in response to infection (Figure 5A,B). GSH was higher in RI than TI at 7 dpi, whereas GSSG was higher in TI than RI at 11 dpi (Figure 5A). Despite this, the GSH/GSSG ratio—a key indicator of cellular redox balance [67,68]—remained consistently higher in RI than in TI from 3 to 11 dpi (Figure 5C), suggesting a more reduced cellular environment (less oxidative damage) in RI during infection.

#### 2.5.7. Flavonoid-Related Pathways

Several DAMs involved in the biosynthesis of flavones/flavonols (Appendix A), flavonoids (Appendix A), phenylpropanoids (PPs; Appendix A), and in the metabolism of Gly, Ser, and Thr (Appendix A) were suppressed in RI. These included quercitrin, kaempferol, quercetin, luteolin, EC, and 5-aminolevulinic acid (Appendix A). In contrast, the isoflavonoid MG increased dramatically in RI, accumulating to levels 5- to 24-fold higher than in RC at 7 and 11 dpi (Figure 4I).

### 2.6. Validating DAM Candidates for Their Potential Roles in Rlm1-Mediated Resistance

Nine metabolites were selected and applied exogenously to the cotyledons of susceptible canola varieties to assess their roles in *Rlm1*-mediated resistance. When applied to Topas and Westar prior to inoculation, all metabolites—along with the phenylpropanoid pathway inhibitor PipA [69] (Table 4)—significantly reduced lesion development compared to controls (*p* < 0.05, LSD; Figure 6). While some metabolites showed slightly greater efficacy in one or both varieties, pipecolic acid (PA) consistently demonstrated the strongest suppression. Additionally, PA also reduced lesion expansion when applied post-inoculation at 1 and 3 dpi on both Topas and Westar but had no effect at 9 dpi (Figure 7).

## 3. Discussion

PCA analysis revealed distinct metabolomic patterns linked to *Rlm1*-mediated resistance, with RI (7 and 11 dpi) and TI (11 dpi) samples clustering separately (Appendix A), suggesting unique metabolic responses associated with *Rlm1* and/or infection duration. Hierarchical clustering heatmaps further supported these distinctions (Figure 3). Several DAMs increased markedly in inoculated Topas–*Rlm1* at 3 dpi and continued to accumulate at 7 and 11 dpi, whereas in inoculated Topas, similar changes were often delayed until 11 dpi. This earlier accumulation may reflect a role in *Rlm1*-mediated resistance.

During RI resistance, multiple pathways were activated, including those related to the biosynthesis of Lys, SA, GA, Met, GABA, GSH, and Mel, as well as Lys degradation (Figure 4). These pathways involved numerous DAMs and are consistent with previous studies linking these metabolites to plant defense. Their specific roles in disease resistance are discussed in the following sections.

Lysine and derivatives: Lys biosynthesis involves key intermediates, such as DAP and meso-DAP [70,71], while its degradation produces cadaverine, glutamic acid, AAA, D-1-P2C, SAP, and ultimately pipecolic acid (PA), a known non-protein amino acid involved in SAR [62,72,73,74]. In this study, both meso-DAP and DAP accumulated significantly in resistant interactions (RI), and PA levels rose 164- and 326-fold at 7 and 11 dpi, respectively, compared to resistant controls (RC) (Figure 4A,B). Lys catabolic intermediates such as SAP, AAA, Nε,Nε-dimethyllysine, N6-Acetyl-Lys, 5-hydroxylysine, and 5-APA also increased significantly in RI (Figure 4A). These findings align with prior studies demonstrating Lys’s role in SAR and its function as a precursor for alkaloid biosynthesis [32,33,75].

Beyond PA and AAA, we observed the accumulation of other amino acids, including GABA, Tyr, and Met—but not Trp—in RI (Figure 4H,P), suggesting broader amino acid pathway involvement in resistance. PA may also coordinate SAR with aspartate-derived amino acid homeostasis (Ile, Met, Thr, Lys) [33]. Similar elevations of PA, AAA, GABA, Tyr, Trp, Leu, Ile, and Lys were previously reported in resistant *Arabidopsis* and tobacco infected with *Pseudomonas syringae* [73,76]. Furthermore, in wheat, conversion of Lys into α-aminoadipate via the SAP pathway contributed to early resistance against *Puccinia striiformis* f. sp. *tritici*, marked by elevated 2-AAA and SAP [77,78].

Exogenous treatments of susceptible canola seedlings with PA, Lys, or DAP significantly limited infection on cotyledons (Figure 6 and Figure 7), supporting their roles in blackleg resistance. Even a quantitative reduction in infection development on cotyledons may restrict pathogen spread into the stem [6]. Notably, PA applications at 1 or 3 dpi also suppressed infection, while treatment at 9 dpi had no effect (Figure 7), indicating that early PA induction—likely during the biotrophic phase—is critical. Once the pathogen enters its necrotrophic phase, PA becomes less effective. This notion is supported by our observation that several DAMs accumulated equally or more in susceptible Topas than in resistant Topas–*Rlm1* by 11 dpi (Figure 3), highlighting the importance of early metabolite induction in *Rlm1*-mediated resistance.

SA: Both free and bound SA levels were highly elevated in RI (Figure 4J), supporting SA’s central role in resistance [79,80,81,82]. Transcriptomic data from Zhai, Liu [17] similarly reported strong activation of SA signaling in *Rlm1*-mediated resistance. SA regulates pathogenesis-related (PR) genes, which encode antimicrobial proteins or amplify defense signaling [83,84,85,86]. Additionally, SA enhances plant defense by promoting callose deposition [87], ROS production [88], and programmed cell death [5,89,90,91].

GA: A secondary metabolite derived from SA [92], GA contributes to plant defense responses, though its role can vary with host–pathogen interactions [93,94]. In this study, both GA and SA significantly accumulated in the resistant interaction (RI) at 7 and 11 dpi, respectively (Figure 4J). Similar to SA, GA can act as a signaling molecule that induces antimicrobial PR proteins, such as P23, P32, and P34 in tomato [93].

However, unlike SA, GA accumulation has been reported primarily in compatible or non-necrotic interactions (e.g., *ToMV*, *CEVd*), but not in incompatible, HR-associated responses [93,95]. For example, in cucumber, low-dose *P. syringae* inoculation triggered GA accumulation during a compatible response, while high-dose inoculation led to HR-like necrosis without GA induction [40]. In contrast, our results show clear GA accumulation during an incompatible interaction with *L. maculans*, suggesting a distinct role for GA in *Rlm1*-mediated resistance in canola.

Exogenous application of GA or SA before inoculation significantly reduced disease symptoms in susceptible Topas (Figure 6), confirming their involvement in defense. SA peaked earlier than GA (7 vs. 11 dpi; Figure 4J), implying a more prominent role for SA during the early, biotrophic phase of infection, with GA potentially contributing during later stages.

Glucosinolates: Several GSL biosynthesis-related compounds, including p-HPAH, Met, and homo-Met, were significantly induced in inoculated Topas–*Rlm1* (Figure 4D). GSL hydrolysis by myrosinase generates antimicrobial products such as isothiocyanates, thiocyanates, and nitriles [96,97,98,99,100]. Met and homo-Met are key intermediates in GSL biosynthesis [101,102,103], and Met also contributes to ethylene production and DNA methylation [104,105,106]. Previous studies showed that Met application or *METS1* overexpression enhanced resistance to rice blast in rice [104,105]. The significant accumulation of Met, homo-Met, and p-HPAH during the incompatible interaction between Topas–*Rlm1* and *L*. *maculans* suggests that the GSL-related pathway plays a role in the resistance.

NAS, Mel, and 5-MT: In *Arabidopsis*, rice, and cassava, NAS is a key precursor in Mel biosynthesis, converted to Mel via *ASMT* [107,108,109], and this activity enhances resistance to *Xanthomonas axonopodis* in benth and cassava [109]. Exogenous application of Mel, NAS, 5-MT, or 5-methoxyindole has similarly boosted resistance in *Arabidopsis*, tobacco, and benth [110,111]. In this study, RI samples showed elevated levels of NAS, 5-MT, and a metabolite (Peak 3618) tentatively identified as Mel (Figure 4F), suggesting potential coordinated up-regulation of Mel biosynthesis during *Rlm1*-mediated resistance.

GABA contributes to plant immunity by acting as a signaling molecule that regulates stress responses, inhibits pathogens directly, and modulates ROS to limit oxidative damage [112,113,114]. In cucumber, exogenous GABA boosts antioxidant enzyme activity, reducing H_2_O_2_ and superoxide levels [114]. It also mediates hormone crosstalk and activates induced systemic resistance (ISR) pathways [112]. In citrus, GABA treatment increased endogenous levels and induced defense-related hormones such as SA, CA, and ABA [115]. The elevated GABA observed in RI canola samples is consistent with its defense-associated roles in other plant–pathogen systems.

GSH, GSSG, and their ratio, along with antioxidant enzymes, are crucial redox components in plant defense against oxidative stress [68]. Exogenous SA enhances GSH levels in tomato [116] and stimulates GSH and chlorogenic acid accumulation in chickpea [117]. During hypersensitive response, GSH and tryptophan-derived metabolites limit pathogen growth in *Arabidopsis* [118]. In wheat, GSSG transport to the apoplast supports class III peroxidase activity, generating ROS for defense against Hessian fly larvae [67], and a high GSH/GSSG ratio is linked to powdery mildew resistance [119]. Consistent with these findings, our results showed induction of both GSH and GSSG in Topas and Topas–*Rlm1* upon blackleg infection, with significantly higher GSH/GSSG ratios in Topas–*Rlm1* (Figure 5), highlighting its role in canola defense.

E-6′-HF is a hydroxy-cinnamic acid (hydroxy-CA) phenolic compound [65], and hydroxy-CAs are well known for their antimicrobial and antioxidant activities [120]. Plants synthesize hydroxy-CA compounds in response to pathogen attack [121]. The accumulation of E-6′-HF, FA, and CFA in Topas–*Rlm1* (Figure 4E & Appendix A) suggests their defensive roles. Phe, via the phenylpropanoid pathway, is a precursor for FA and CFA, which in turn contribute to E-6′-HF biosynthesis [65,66]. FA, derived from cinnamic acid (CA), activates the phenylpropanoid pathway and stimulates ROS production, enhancing plant defense [122]. This phenylpropanoid pathway also leads to the synthesis of SA, phytoalexins, and lignin, key components of plant immunity [123].

CFA derivatives reinforce plant cell walls and enhance defense, also exhibiting direct antimicrobial effects—for example, nanoparticle formulations with methyl-CFA and CFA-phenethyl-ester effectively target *Ralstonia solanacearum* [124,125,126]. Unexpectedly, inoculated Topas–*Rlm1* showed no significant increase in Phe levels compared to susceptible Topas (Figure 4E and Appendix A), despite Phe being the precursor for FA and CFA in the phenylpropanoid pathway and commonly associated with disease resistance [127,128,129]. This suggests that FA and CFA biosynthesis in Topas–*Rlm1* may be regulated independently of Phe availability, potentially via pathway modulation during infection.

In this study, peak 3324 (Figure 4N) tentatively matched both IAA and 5-HIAL in the Tier 3 library, suggesting involvement in Trp metabolism and IAA biosynthesis [63,64]. While Trp is a known precursor for several defense-related metabolites [63,64], its levels did not increase in inoculated Topas–*Rlm1*. Instead, peak 3324, 2-AA, 2-AP, and 5-HKA accumulated at 7 and/or 11 dpi (Figure 4N). 2-AA, a key IAA intermediate, is also linked to SA-dependent ISR and PR gene activation [130,131,132,133], while 2-AP and 2-AA have antimicrobial properties [134]. Notably, 5-HKA has been associated with resistance in soybean roots during *Phytophthora sojae* infection [135], supporting its potential role in canola defense.

Several metabolites linked to ACN biosynthesis and plant defense accumulated significantly in inoculated Topas–*Rlm1* plants (Figure 4C,I,K), though their specific roles in resistance remain unclear. In grapevine, ACN pathway metabolites have been associated with SAR against *Botrytis cinerea* [136]. Notably, NCP, a polyamine biosynthesis intermediate [137,138], accumulated early in Topas–*Rlm1* but increased more slowly than in Topas (Figure 4K). NCP contributes to spermine production [137,138], which enhances resistance by triggering HR and SA-independent PR proteins [139,140]. Additionally, the isoflavonoid MG increased in Topas–*Rlm1* (Figure 4I), consistent with its role in soybean resistance to *Euschistus heros* [52], suggesting a potential defense function in canola.

Exogenous application of selected DAMs—including PA, FA, CFA, SA, GA, DAP, GSH, BA, and Lys, as well as the inhibitor of the phenylpropanoid pathway PipA [69]—significantly reduced infection in susceptible Topas and/or Westar (Figure 6 and Figure 7), confirming their roles in blackleg resistance. Repeated applications before inoculation likely maximized treatment effects, while early post-inoculation application of PA also proved effective, emphasizing the importance of early activation of these metabolites (Figure 7).

Several DAMs with known or potential antimicrobial activity were enriched in inoculated Topas–*Rlm1* plants, though their timing varied (Figure 4K). Medicarpin, PZD, and mangiferin increased early (3 dpi), while 3-HMA, EMD, and PG accumulated later. Medicarpin, a phytoalexin, activates SA-related defense pathways and has been linked to powdery mildew resistance in *Medicago truncatula* [45,46]. Mangiferin, PZD, and PG derivatives also exhibit antimicrobial properties, with mangiferin shown to inhibit *Fusarium oxysporum* in vitro and in planta [47,48,49]. Similarly, 3-HMA suppresses spore germination and hyphal growth of *F. oxysporum* [50]. EMD, which modestly increased at 7 dpi, has been associated with phytoalexin induction and HR [51]. While these compounds may contribute to defense, their roles in *Rlm1*-mediated resistance have not been confirmed.

Conversely, several DAMs involved in phenylpropanoid, flavonoid, and Gly/Ser/Thr metabolism showed reduced accumulation in *Rlm1* plants (Figure 4Q and Appendix A). Although linked to resistance or stress tolerance in other species [58,141,142,143], their down-regulation here suggests that these metabolites may be non-essential or even detrimental in the context of blackleg resistance. Some phenolic compounds can facilitate pathogen colonization [144], and selective suppression of these may prevent exploitation by the pathogen. Metabolic flux from the phenylpropanoid pathway can also be redirected toward SA biosynthesis [145,146], a key defense signal [147]. Supporting this, treatment with PipA—a phenylpropanoid pathway inhibitor [69]—reduced disease symptoms in susceptible Topas (Figure 7B), suggesting that blocking certain branches enhances SA-mediated resistance [69,145,146].

While this study provides valuable insights into metabolite profiles associated with *Rlm1*-mediated resistance to *L*. *maculans*, several limitations should be noted. First, the untargeted metabolomics approach enabled broad DAM identification, but definitive confirmation—particularly of Tier-3 compounds—was limited by the absence of authentic standards or MS/MS data. Second, the controlled growth-incubator conditions may not fully replicate field environments, where abiotic stressors, soil microbiota, and variable pathogen pressures could influence metabolite responses. Third, although some DAMs suppressed infection when applied exogenously, the underlying mechanisms remain undetermined, and such applications may not reflect endogenous biosynthesis or localization exactly. Lastly, the exclusive use of Topas and Topas–*Rlm1* may not capture responses mediated by additional R genes present in commercial cultivars.

Despite these constraints, this study offers a foundation for future research. Targeted metabolomics with authentic standards and MS/MS validation should be used to confirm and quantify key DAMs, particularly those putatively annotated but shown to have strong associations with resistance, such as NAS, Mel, 5-MT, and 2-AA. Integrating transcriptomic and proteomic data will help elucidate gene–metabolite networks and regulatory mechanisms. Functional validation of candidate biosynthetic genes through CRISPR/Cas9, via overexpression or knockout, can establish causal links to resistance. Finally, extending analyses to other R genes or QTLs may uncover both gene-specific and broad-spectrum metabolic signatures, advancing efforts toward durable blackleg resistance.

## 4. Materials and Methods

### 4.1. Plant Materials and Pathogen Isolates

A near isogenic line (NIL) of *B*. *napus* carrying the *R* gene *Rlm1* was developed at the Saskatoon Research and Development Center, AAFC [148]. The process used the ‘Quinta’ double haploid (DH) line DH24288, which carries *Rlm1* and *Rlm3* [149], as the *R* parent in backcrossing with the susceptible Topas DH16516 [148]. The NIL and Topas DH16516 were used as resistant and control lines, respectively, throughout the study. Each line was planted in 128-well flats filled with Sunshine #3 soilless mix (Sun Gro Hort. Canada Ltd., Vancouver, BC, Canada) amended with 12.5 g L^−1^ Osmocote Plus 16-9-12 (N-P-K, Scotts Miracle-Gro Canada Ltd., Mississauga, ON, Canada). Seeded flats were placed in an incubator set with a day/night temperature regime of 22/18 °C, about 65% relative humidity, and a daily photoperiod (427 µmol·m^−2^·s^−1^) of 16 h. In later experiments, to validate the effect of putative metabolites identified on blackleg resistance, a DH line of ‘Westar’ [150,151] was planted similarly as an additional susceptible control.

The *L. maculans* isolate Sc006 carrying the avirulence gene *AvrLm1* was used to inoculate all plants. The isolate was cultured on V8-juice agar amended with streptomycin sulfate (100 ppm) at 20 °C under cool-white fluorescent light (325 µmol·m^−2^·s^−1^) for 7–10 d for inoculum production [152]. Pycnidiospores were harvested by flooding the culture plates with sterilized water and filtering resulting spore suspension through a Falcon™ Cell Strainer (70 μm, Corning/Sigma-Aldrich, Markham, ON, Canada). The concentration of the obtained spore suspension was estimated using a hemocytometer and adjusted to 2 × 10^7^ spores/mL with sterilized water for plant inoculation.

### 4.2. Plant Inoculation, Infection Assessment, and Leaf-Tissue Sampling

About a week after seedling emergence, each lobe of cotyledon was pricked with a pair of bent-tipped tweezers, and each wound was inoculated with a 10-μL droplet of prepared spore suspension. Wounds receiving sterilized water were used as non-inoculated controls (mock). Inoculated seedlings were air dried at room temperature for 30 min before being placed back in the incubator. Following the inoculation, emerging true leaves were removed every 3–5 days to delay the senescence of inoculated cotyledons. At 12–14 days post-inoculation (dpi), the severity of infection on inoculated and control cotyledons was assessed using a 0–9 scale introduced by Koch, Badawy [153], with cotyledon tissues immediately around the inoculation site or expanding lesion being sampled using a paper puncher at 3, 7, and 11 dpi, respectively. This hemibiotrophic fungus tends to establish a transient biotrophic relationship with the host following successful infection (3 dpi) before transitioning to the necrotrophic phase around 7–10 dpi, when visible necrotic lesions begin to form [17,154]. Samples from 20 random seedlings of the same treatment were bulked and grounded in liquid nitrogen into a fine powder using a mortar and pestle to form a biological replicate, with three replicates prepared for each treatment or control at each of the time points of sampling. All bulked samples were stored at −80 °C until use.

### 4.3. Sample Preparation for Metabolomic Analysis Using CIL LC–MS

Bulked cotyledon samples were extracted for metabolites following the protocol described by Tunsagool, Wang [31] at the Metabolomics Innovation Centre (TMIC), University of Alberta, with only slight modifications. Chemicals and reagents were sourced from Sigma-Aldrich (Markham, ON, Canada) unless indicated otherwise. Briefly, 300 mg of each sample was transferred into a 2 mL Eppendorf tube containing 1.5 mL of extraction buffer (methanol/water, 4:1, *v*/*v*) and 2.8 mm ceramic beads. After vortexing for 15 s, the sample was homogenized using the Bioprep-24 homogenizer (Allsheng, Hangzhou, China) for another 15 s. The tube was then placed in a −20 °C freezer for 10 min before being centrifuged at 15,000× *g* at 4 °C for 10 min (Eppendorf 5430R, Mississauga, ON, Canada). The resulting supernatant was transferred to a fresh vial for subsequent metabolomic analysis.

Dansylation labeling, qualification, and LC–MS analysis of samples generally followed the protocol described previously by Luo, Zhao [29]; 25 µL of each extraction was dried under nitrogen blowdown and reconstituted in an equal volume of LC–MS grade water (Canadian Life Sciences, Peterborough, ON, Canada). The samples were then processed following the SOPs provided in the kit by the manufacturer of Dansyl-labeling Kit for Amine & Phenol Metabolomics (NMT-4101-KT, Nova Medical Testing Inc., Edmonton, AB, Canada). Each sample was labeled using ^12^C-labeling reagent. To establish a reference standard, a pooled sample was produced by mixing an equal amount of extraction from each individual sample, labeled with a ^13^C-labeling agent, as a baseline for metabolomic analysis [29].

### 4.4. Metabolome Quantification

An LC–UV-based standard operating protocol (SOP) established at TMIC was used to quantify labeled metabolites for sample amount normalization [155]. All dansylation-labeled samples were centrifuged at 15,000× *g* for 10 min, with 25 µL of each supernatant being used for UV quantification in an HPLC vial (Agilent, Boulder, CO, USA). Each ^12^C-labeled sample was mixed with an equal molar amount of the ^13^C-labeled pooled sample based on the SOP. This mixture was measured using LC–MS analysis for the peak intensity ratio of metabolites between individual samples and the pooled sample [29]. Quality control (QC) over the LC–MS analysis was performed by combining equal amounts of ^12^C-labeled and ^13^C-labeled pools.

### 4.5. LC−MS Analysis

LC–MS analysis of labeled metabolite samples was conducted following the previously reported method [42] at TIMC, utilizing a Thermo Vanquish UHPLC system coupled to a Q Exactive Orbitrap mass spectrometer (Thermo Scientific, Edmonton, AB, Canada). To ensure consistent instrument performance, quality control (QC) samples were injected after every 20 sample runs. Mixed ^12^C- and ^13^C-labeled samples were separated using an Eclipse Plus C18 reversed-phase column (2.1 mm × 15 cm, 1.8 μm particle size; Agilent Technologies., Mississauga, ON, Canada), with mobile phases and gradient conditions adapted from Luo, Zhao [29]. Solvent A consisted of 0.1% (*v*/*v*) formic acid in water, and solvent B was 0.1% (*v*/*v*) formic acid in acetonitrile. The gradient program was 25% B at 0 min, ramped to 99% B by 10 min, held at 99% B until 13 min, returned to 25% B at 13.1 min, and maintained until 16 min. MS conditions were as follows: flow rate of 400 μL/min, column temperature at 40 °C, mass range *m*/*z* 220–1000, and acquisition rate of 1 Hz. The gradient and mass range were adopted from a well-established protocol specifically for chemical isotope labeling LC–MS metabolomics [42], which optimize LC–MS parameters for metabolites labeled with dansyl chloride by increasing both mass and hydrophobicity of metabolites.

### 4.6. LC–MS Raw Data Extraction and Processing

The open-source ProteoWizard MSConvert software v 2.1 (https://proteowizard.sourceforge.io/; accessed on 2 July 2022) was used to convert all LC–MS raw data from profile mode into .txt files [156], and a suite of programs developed at TIMC was employed to process the converted data in batch mode [157]. IsoMS Pro 1.2.5 software [157] was used to pick and align peak pairs and calculate the intensity ratio of metabolites. Peak pairs that were not present in at least 80.0% of samples in any group were filtered out. Data were then normalized by Ratio of Total Useful Signal, calculated as the sum of ^12^C-labeled peak signals divided by the sum of ^13^C-labeled peak signals [158]. This ratio served as a post-acquisition normalization method [158].

For metabolite identification, three-tiered databases developed at TMIC were utilized, including the labeled standard database (CIL Library) as Tier-1 [41], the linked identity library (LI Library) as Tier-2 [42], and MyCompoundID library as Tier-3 [43]. Tier-1 database provided positive identification results by matching to experiential information of compound standard, with high-confidence identification. Tier-2 delivered high-confidence putative identification results based on both experimental and predicted information. Tier-3 generated putative results in which peaks with multiple matches would not be considered. For compounds with multiple matching peaks, only the peak with the lowest absolute error in mass and retention time was retained for analysis. Extracted LC–MS peak data are presented in Appendix A.

### 4.7. Validating the Potential Involvement of Selected Metabolites in Resistance

#### 4.7.1. Chemical (Metabolite) Preparation

Table 4 lists the most prominent DAMs and related pathways identified in inoculated *Rlm1* plants, including PA, SA, GA, GSH, Lys, and DAP. Several DAMs found in non-inoculated *Rlm1* plants, including FA and CFA, were also selected for functional validation due to their high accumulations. The ferulic acid E-6′-HF, along with FA and CFA, are involved in scopoletin (SCF)/isoscopoletin (IsoScp) biosynthesis [65,66]. However, E-6′-HF is unavailable commercially; therefore, only CFA and trans-FA (trans-ferulic acid, a ferulate isomer in plants) were tested in this experiment. Although BA (peak 1365) was only tentatively identified based on Tier-3 database annotation, it was selected for this investigation due to its notably high levels in non-inoculated *Rlm1* plants. PipA is a well-known inhibitor of the phenylpropanoid pathway in plants [69], and may play a role in this context, as several DAMs associated with this pathway were suppressed during *Rlm1*-mediated incompatible interaction. However, for most of these metabolites, their roles in blackleg resistance remain uncharacterized and have yet to be experimentally validated.

Most of the selected metabolites could be dissolved in water, while FA and PipA had to be dissolved first in methyl sulfoxide (DMSO, Sigma-Aldrich Canada, Oakville, ON, Canada) before being diluted to desired concentrations using deionized water. The final solution of FA and PipA contained 0.17% and 1.7% of DMSO, both at low enough concentrations to have minimum effects on canola seedlings. CFA and BA were dissolved initially in 95% ethanol, then diluted with water to achieve desired concentrations in which the ethanol content was 9.5% and 6.8%, respectively. All final metabolite preparations were amended with Triton X-100 (surfactant) at 0.01% for improved spreadability and adherence during spray applications. Water, 0.17%/1.7% of DMSO, or 9.5%/6.8% of ethanol amended with the surfactant was used as a control depending on the treatment.

#### 4.7.2. Application of Metabolites

Two canola varieties, Topas (susceptible) and Westar (highly susceptible), were used to validate the effects of the metabolites. These DAM preparations were applied to canola seedlings using a misting bottle (Uline, Milton, ON, Canada) twice daily, with approximately 0.3 mL per seedling per application, starting two days after emergence and continuing for five consecutive days until the day of inoculation. In addition to the pre-inoculation effect, PA was also studied for its post-inoculation effect due to its highest efficacy of infection suppression observed, with twice-daily spray applications at 1, 3, and 9 dpi.

#### 4.7.3. Inoculation and Infection Assessment

About one hour after the final DAM treatment, each cotyledon lobe was inoculated with the highly virulent *L*. *maculans* isolate (Sc006) and the infection development was assessed using the 0–9 scale at 14 dpi [153], as described above. Inoculated cotyledons sprayed with water, or with the corresponding low concentrations of DMSO or ethanol solution at matched time points, served as controls (Table 4). The experiments followed a completely randomized design (CRD). All metabolites were evaluated in 2–3 independent trials, with 4 to 24 biological replicates (plants) per treatment per trial, depending on the metabolite and trial.

### 4.8. Data Analysis

Most data analyses were performed using R (ver.4.3.3) [159] and RStudio (ver.2023.12.1) [160]. Metabolomics data were processed initially with a 10^4^ integer transformation prior to normalization. Multivariate and univariate analyses were performed using the DESeq2 package [161], which applies the Benjamini–Hochberg method to correct for multiple testing, to explore overall data structure/patterns/group separation across samples (multivariate), and to identify individual metabolites that significantly contribute to these differences (univariate). This integrated approach allows us to gain both global (pattern-level) and specific (feature-level) understanding of the data. Fold changes (FC) were calculated for sample pairs at 3, 7, and 11 dpi. Metabolites were classified as differentially accumulated if log2FC ≥|1| and false discovery rate (FDR)–adjusted *p* ≤ 0.05. Principal component analysis (PCA) was carried out with the “prcomp” function from the base R package [159] and visualized with the “ggplot2” package [162,163,164]. Venn diagrams were created using the “ggvenn” package [165] to illustrate the overlap or uniqueness of metabolites among different treatment groups or time points, and heatmap analysis was performed with the “pheatmap” package [166] to provide a detailed visualization of the abundance levels of metabolites across samples or conditions.

Additionally, the relative intensities of BA, GSH, and oxidized glutathione (GSSG) in the LC–MS chromatogram, as well as the ratio of GSH/GSSG readings, were analyzed initially with ANOVA using “rstatix” [167]; a reduced ratio is a key indicator of cellular redox balance, which is critically relevant to plant disease resistance [68]. The post hoc examination of treatment means used LSD in the “agricolae” [168], visualized with “ggplot2” [162,163,164]. Pathway analysis of highly regulated DAMs was performed using the Arabidopsis metabolite database in MetaboAnalyst6.0 [44].

All data on infection severity of inoculated cotyledons from repeated trials were pooled due to general homogeneity of variance. Pooled data were analyzed for normality (Shapiro–Wilk test) prior to ANOVA, followed with LSD post hoc analysis for mean separation when ANOVA showed significance (*p* ≤ 0.05). For data that lacked a normal distribution, aligned-ranks transformation ANOVA (ART ANOVA) was performed using the “ARTool” package [169,170,171,172], followed with a post hoc test using “emmeans” [173]. Significance groupings were labeled using the “rcompanion” package based on *p*-values [170,174].

## 5. Conclusions

Using the susceptible Topas and its isogenic *Rlm1*-carrying resistant counterpart, combined with the CIL LC–MS platform for metabolomic analysis, we identified distinct metabolic profiles associated with compatible and incompatible interactions during infection by *L*. *maculans*. These differences were most pronounced at specific post-inoculation time points. Our hypotheses regarding the metabolic divergence between susceptible and resistant canola—and the suitability of the CIL LC–MS platform for this investigation—were largely validated. As a result, our objectives were successfully met.

In Topas–*Rlm1*, several key defense-related metabolites were strongly and consistently up-regulated, most notably PA, which increased by up to 326-fold, underscoring its potential central role in resistance. Elevated levels of SA, its derivative GA, glucosinolate pathway intermediates, and GABA further suggest a coordinated hormonal and antimicrobial response. Conversely, lower activity in flavonoid and amino acid metabolism in the resistant line implies these pathways may be less critical to *Rlm1*-mediated defense. Moreover, exogenous application of several defense-associated metabolites—including PA, SA, and GA—significantly reduced infection in susceptible canola varieties, supporting their functional role in blackleg resistance. Overall, this study highlights key metabolic components underpinning *Rlm1*-mediated resistance and demonstrates the potential of metabolite-based approaches—including metabolic engineering—as promising strategies for enhancing blackleg resistance in canola.

## Figures and Tables

**Figure 1 ijms-26-05627-f001:**
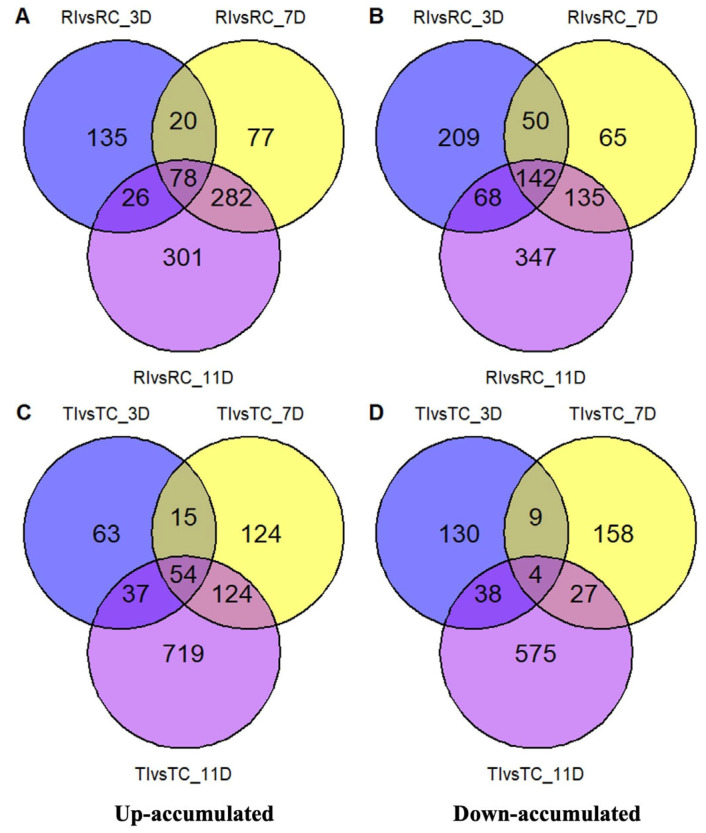
The number of differentially up- (**A**,**C**) and down- (**B**,**D**) accumulated metabolites (DAMs) in inoculated (I) and control (C) samples of Topas (T) and Topas–*Rlm1* (R) within the same treatment collected at 3, 7, and 11 days post-inoculation (dpi). RIvsRC_3D represents the number of DAMs between inoculated and control Topas–*Rlm1* samples collected at 3 dpi. Similarly, TIvsTC_11D shows inoculated and control Topas collected at 11 dpi, and so on.

**Figure 2 ijms-26-05627-f002:**
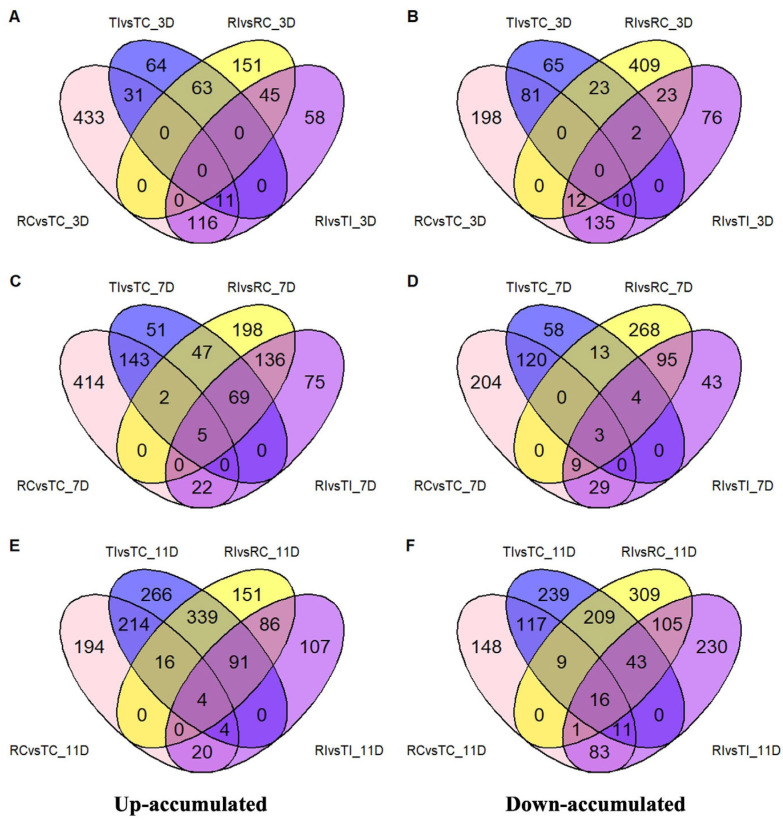
The number of significantly up- (**A**,**C**,**E**) and down-regulated (**B**,**D**,**F**) DAMs among samples between different treatments collected at 3, 7, and 11 dpi.

**Figure 3 ijms-26-05627-f003:**
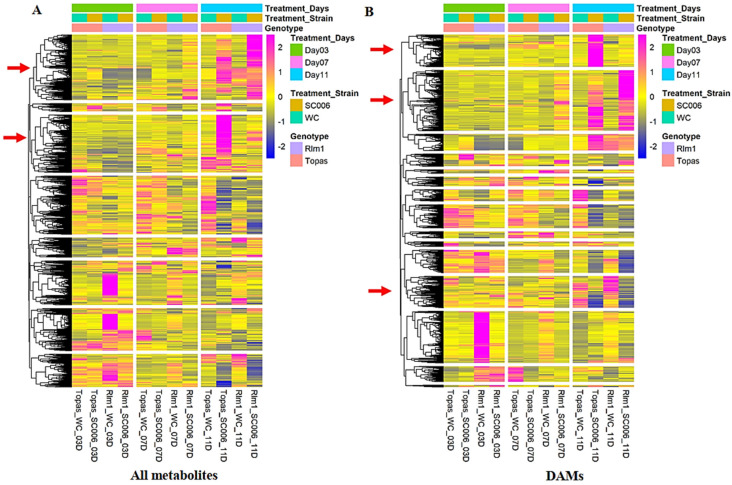
Heatmap clustering analysis of Topas–*Rlm1* and Topas at different stages of infection over (**A**) all metabolites and (**B**) differentially accumulated metabolites (DAMs). Red and blue colors in scale bars represent up- and down-regulated metabolomes, respectively. Red arrows point to the clusters with significantly higher or suppressed accumulation in later stages of infection (7 and/or 11 dpi). Treatment names are consistent with those in Figure 1.

**Figure 4 ijms-26-05627-f004:**
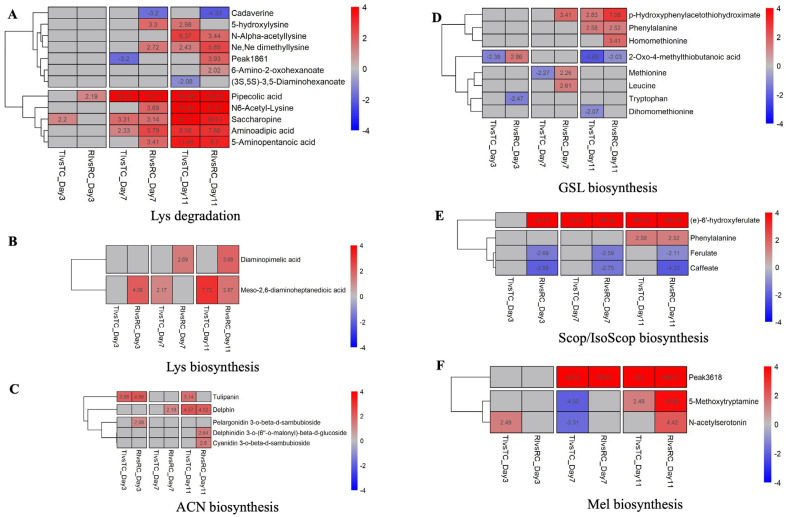
Differentially accumulated metabolites associated with *Rlm1*-mediated resistance that are involved in: (**A**) degradation of lysine (Lys), (**B**–**F**) biosynthesis of Lys, anthocyanin (ACN), glucosinolate (GSLs), scopoletin/isoscp (Scop/IsoScop), and melatonin (Mel), (**G**,**H**) metabolism of nicotinate-nicotinamide (NaN) and γ-aminobutyric acid (GABA), (**I**,**J**) defense-related isoflavonoids (Isoflav) and hormones, (**K**) and antimicrobial/plant-defense responses [45,46,47,48,49,50,51,52]. (**L**) Biosynthesis of tropane (Trop), piperidine (Pid), and pyridine alkaloid. (**M**) Biosynthesis of phenylalanine (Phe), tyrosine (Tyr), and tryptophan (Trp). (**N**,**O**) Metabolism of Trp, taurine (Tau)/hypotaurine (HTau). (**P**) Other amino acids and their derivatives. (**Q**) Metabolism of Phe. (**R**) The glutamate family. (**S**) Response to abiotic stress [53,54,55,56,57,58,59,60]. Red and blue colors represent up- and down-regulated DAMs, respectively, while gray color indicates no changes from controls. Treatment names are consistent with those designated in Figure 1.

**Figure 5 ijms-26-05627-f005:**
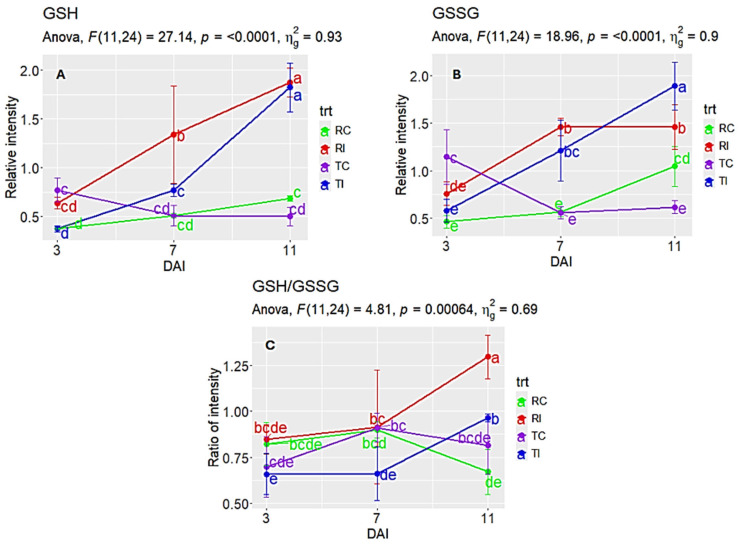
Relative intensities of glutathione (GSH) and oxidized glutathione (GSSG) in LC–MS analysis, as well as their ratios, in Topas (T) and Topas–*Rlm1* (R) canola receiving *L*. *maculans* (I) or water (C). (**A**,**B**) LC–MS intensity for GSH and GSSG. (**C**) The ratio of GSH/GSSG for each treatment. Data points with the same letter(s) across DAI (day after inoculation) did not differ significantly (*p* > 0.05, LSD).

**Figure 6 ijms-26-05627-f006:**
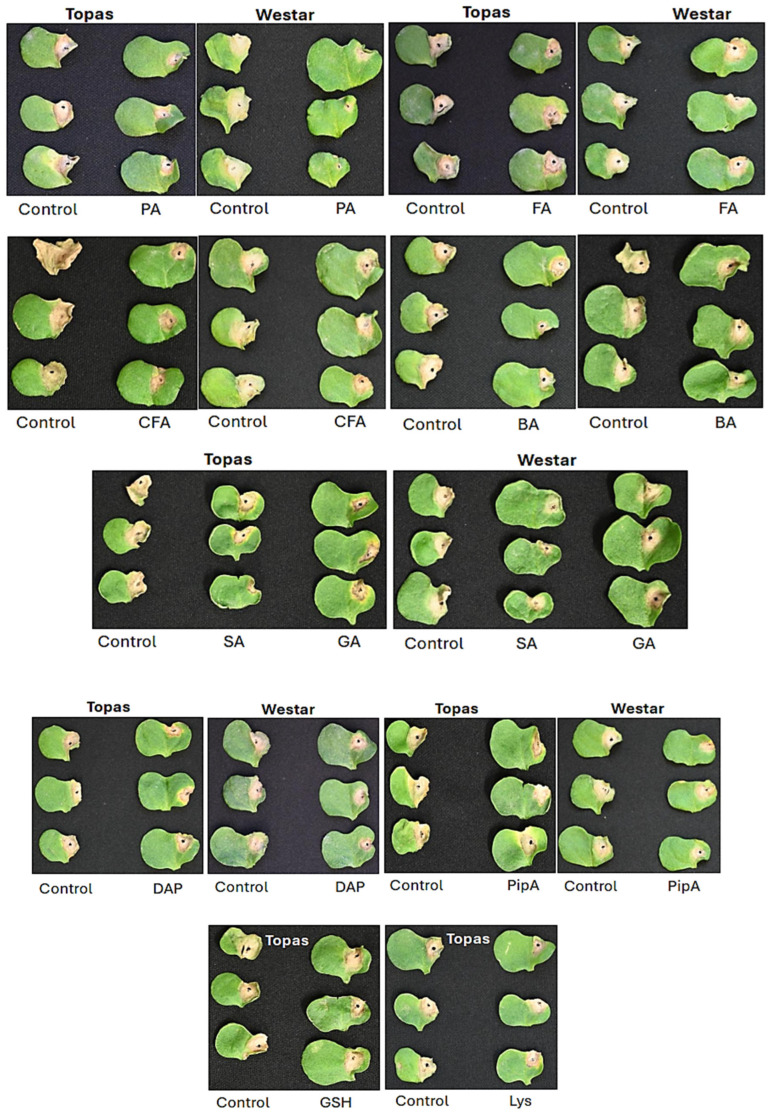
Suppression of *L*. *maculans* infection on cotyledons of Topas (moderately susceptible) and Westar (highly susceptible) following treatment with various metabolites. Treatments included pipecolic acid (PA, 40 mM), ferulic acid (FA, 1 mM), caffeic acid (CFA, 10 mM), benzoic acid (BA, 10 mM), salicylic acid (SA, 1 mM), gentisic acid (GA, 10 mM), 2,6-diaminopimelic acid (DAP, 30 mM), and piperonylic acid (PipA, 3 mM) applied prior to inoculation. Infection severity was assessed at 14 days post-inoculation. All treatments reduced infection compared to the control on both canola varieties (*p* < 0.05, LSD), except glutathione (GSH, 20 mM) and lysine (Lys, 10 mM), which showed efficacy only on Topas. Three inoculated cotyledons were photographed for each treatment to illustrate the range of symptoms observed.

**Figure 7 ijms-26-05627-f007:**
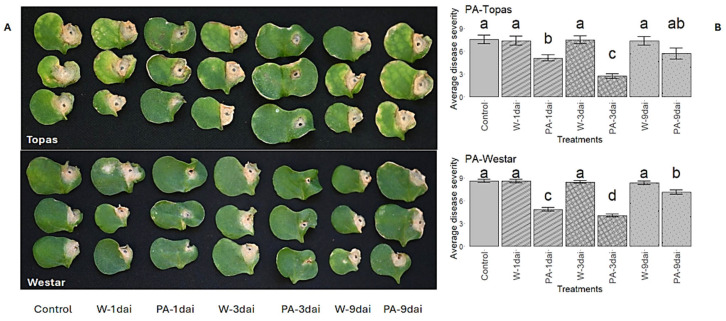
Effect of post-inoculation treatments with pipecolic acid (PA, 40 mM) on infection of Topas and Westar cotyledons inoculated with *L*. *maculans*. (**A**) Symptoms at 14 dai (days after inoculation). Three inoculated cotyledons were photographed for each treatment to show the range of symptoms observed. (**B**) The mean infection severity where treatments with different letters are significantly different (*p* < 0.05, LSD).

**Table 1 ijms-26-05627-t001:** Pathways involved in *Rlm1*-mediated resistance at 3 dpi ^1^.

Pathways	Total Compounds	Hits	Raw *p*	Impact
Flavone and flavonol biosynthesis	10	4	5.2649 × 10^−5^	0.5
Isoquinoline alkaloid biosynthesis	6	1	0.013614	0.5
Arginine and proline metabolism	32	3	0.11938	0.32738
Biosynthesis of various plant secondary metabolites	29	1	0.24485	0.24
Glycine, serine and threonine metabolism	33	2	0.018356	0.22375
Lysine biosynthesis	9	1	0.000103	0.16216
Arginine biosynthesis	18	1	0.17605	0.13981
Tryptophan metabolism	29	1	0.010177	0.10687
Glyoxylate and dicarboxylate metabolism	29	1	0.11936	0.10147
Tyrosine metabolism	17	1	0.013614	0.10056
Phenylpropanoid biosynthesis	43	2	0.00128	0.09634
Glutathione metabolism	26	1	0.11936	0.07114
Pyrimidine metabolism	41	1	0.021007	0.02929
Purine metabolism	73	3	0.000316	0.02344
Phenylalanine, tyrosine, and tryptophan biosynthesis	22	2	0.011081	0.02002
Flavonoid biosynthesis	47	2	4.2424 × 10^−5^	0.00338
Cysteine and methionine metabolism	47	1	0.000562	0.00265
Lipoic acid metabolism	24	1	0.11936	0.0016
D-Amino acid metabolism	7	1	0.000103	0
Indole alkaloid biosynthesis	4	1	0.010177	0
Glucosinolate biosynthesis	65	1	0.010177	0
Ubiquinone and other terpenoid-quinone biosynthesis	47	1	0.013614	0
Anthocyanin biosynthesis	11	1	0.014562	0
Lysine degradation	20	1	0.038832	0
Thiamine metabolism	22	1	0.11936	0
Cyanoamino acid metabolism	29	2	0.16748	0

^1^ Pathway enrichment analysis was conducted using the MetaboAnalyst6.0 (https://www.metaboanalyst.ca; accessed on 10 October 2024) on DAMs identified in RI at 3 dpi [44]. ‘Total compounds’ refers to all potential metabolites associated with the pathway based on the MetaboAnalyst6.0 database. ‘Hits’ indicates the number of DAMs detected in the sample for the pathway. ‘Raw *p*’ represents unadjusted *p*-values from the pathway enrichment analysis. ‘Impact’ denotes the influence of DAMs on the pathway based on the enrichment analysis.

**Table 2 ijms-26-05627-t002:** Pathways involved in *Rlm1*-mediated resistance at 7 dpi ^1^.

Pathways	Total Compounds	Hits	Raw *p*	Impact
Taurine and hypotaurine metabolism	5	2	0.004092	1
Glutathione metabolism	26	5	0.000843	0.51637
Isoquinoline alkaloid biosynthesis	6	3	0.000207	0.5
Tyrosine metabolism	17	4	0.001364	0.32961
Glycine, serine and threonine metabolism	33	1	7.41 × 10^−5^	0.22375
Arginine biosynthesis	18	2	0.020249	0.17088
Lysine degradation	20	3	5.38 × 10^−5^	0.16667
Lysine biosynthesis	9	1	0.000818	0.16216
Flavone and flavonol biosynthesis	10	2	0.00029	0.15
Arginine and proline metabolism	32	3	0.040159	0.14584
Butanoate metabolism	17	1	0.13621	0.13636
Cysteine and methionine metabolism	47	2	0.000311	0.13181
Alanine, aspartate, and glutamate metabolism	22	1	0.13621	0.1295
Purine metabolism	73	3	0.001022	0.10374
Glyoxylate and dicarboxylate metabolism	29	1	7.41 × 10^−5^	0.10147
Phenylpropanoid biosynthesis	43	1	0.017894	0.05935
Pyrimidine metabolism	41	1	0.000913	0.02929
Folate biosynthesis	31	1	0.39026	0.02624
Porphyrin metabolism	48	1	0.001198	0.02261
Ubiquinone and other terpenoid-quinone biosynthesis	47	2	4.84 × 10^−5^	0.02209
Phenylalanine, tyrosine, and tryptophan biosynthesis	22	2	4.84 × 10^−5^	0.02002
Tryptophan metabolism	29	3	0.000722	0.01527
Lipoic acid metabolism	24	1	7.41 × 10^−5^	0.0016
Thiamine metabolism	22	1	7.41 × 10^−5^	0
Cyanoamino acid metabolism	29	2	8.89 × 10^−5^	0
Glucosinolate biosynthesis	65	3	0.000128	0
Flavonoid biosynthesis	47	1	0.000193	0
Biosynthesis of various plant secondary metabolites	29	1	0.000248	0
Valine, leucine, and isoleucine degradation	37	1	0.000662	0
Valine, leucine, and isoleucine biosynthesis	22	1	0.000662	0
D-Amino acid metabolism	7	1	0.000818	0
Tropane, piperidine, and pyridine alkaloid biosynthesis	9	2	0.003453	0
Anthocyanin biosynthesis	11	1	0.012109	0
Zeatin biosynthesis	21	1	0.044967	0

^1^ Pathway enrichment analysis was conducted using the MetaboAnalyst6.0 (https://www.metaboanalyst.ca; accessed on 10 October 2024) on DAMs identified in RI at 7 dpi [44]. ‘Total compounds’ refers to all potential metabolites associated with the pathway based on the MetaboAnalyst6.0 database. ‘Hits’ indicates the number of DAMs detected in the sample for the pathway. ‘Raw *p*’ represents unadjusted *p*-values from the pathway enrichment analysis. ‘Impact’ denotes the influence of DAMs on the pathway based on the enrichment analysis.

**Table 3 ijms-26-05627-t003:** Pathways involved in *Rlm1*-mediated resistance at 11 dpi ^1^.

Pathways	Total Compounds	Hits	Raw *p*	Impact
Taurine and hypotaurine metabolism	5	3	0.003645	1
Phenylalanine metabolism	12	1	0.000113	0.42308
Glutathione metabolism	26	3	9.13 × 10^−5^	0.40276
Tyrosine metabolism	17	5	0.000141	0.39665
Phenylpropanoid biosynthesis	43	8	9.23 × 10^−5^	0.28583
Ubiquinone and other terpenoid-quinone biosynthesis	47	2	0.000443	0.1998
beta-Alanine metabolism	18	2	0.022568	0.19444
Lysine degradation	20	3	0.000743	0.16667
Lysine biosynthesis	9	1	0.000192	0.16216
Arginine and proline metabolism	32	3	3.65 × 10^−6^	0.15774
Butanoate metabolism	17	1	0.000299	0.13636
Alanine, aspartate and glutamate metabolism	22	1	0.000299	0.1295
Purine metabolism	73	3	0.003473	0.09255
Phenylalanine, tyrosine and tryptophan biosynthesis	22	3	1.11 × 10^−5^	0.09159
Arginine biosynthesis	18	1	0.004743	0.08641
Pyrimidine metabolism	41	2	0.002791	0.07198
Flavonoid biosynthesis	47	5	0.00363	0.06956
Cysteine and methionine metabolism	47	5	0.000575	0.05644
Glucosinolate biosynthesis	65	3	1.36 × 10^−6^	0.04236
Tryptophan metabolism	29	4	4.93 × 10^−5^	0.03054
Pantothenate and CoA biosynthesis	25	1	0.16968	0.02796
Porphyrin metabolism	48	1	0.013372	0.02261
Flavone and flavonol biosynthesis	10	2	6.24 × 10^−5^	0
Cyanoamino acid metabolism	29	1	0.000113	0
Tropane, piperidine and pyridine alkaloid biosynthesis	9	3	0.000161	0
D-Amino acid metabolism	7	1	0.000192	0
Anthocyanin biosynthesis	11	2	0.00027	0
Glycine, serine and threonine metabolism	33	1	0.002717	0
Zeatin biosynthesis	21	1	0.005646	0
Isoquinoline alkaloid biosynthesis	6	2	0.006486	0

^1^ Pathway enrichment analysis was conducted using the MetaboAnalyst6.0 (https://www.metaboanalyst.ca; accessed on 10 October 2024) on DAMs identified in RI at 11 dpi [44]. ‘Total compounds’ refers to all potential metabolites associated with the pathway based on the MetaboAnalyst6.0 database. ‘Hits’ indicates the number of DAMs detected in the sample for the pathway. ‘Raw *p*’ represents unadjusted *p*-values from the pathway enrichment analysis. ‘Impact’ denotes the influence of DAMs on the pathway based on the enrichment analysis.

**Table 4 ijms-26-05627-t004:** Most differentially accumulated metabolites/hormones used to validate their involvement in *Rlm1*-mediated resistance to *L*. *maculans* in canola ^1^.

Common Name	Abbreviation	Chemical Name	Mol. Formula	Concentration ^2^	Supplier
Pipecolic acid	PA	Piperidine-2-carboxylic acid	C_6_H_11_NO_2_	40 mM	Tokyo Chemical Industry (TCI) (Portland, OR, USA)
Salicylic acid (sodium salt)	SA	Sodium 2-hydroxybenzoate	C_7_H_5_NaO_3_	1 mM	Thermo Fisher (Ottawa, ON, Canada)
Gentisic acid (sodium salt hydrate)	GA	2,5-Dihydroxybenzoic acid sodium salt	C_7_H_5_O_4_Na	10 mM	Sigma Aldrich (MilliporeSigma Canada Ltd., Oakville, ON, Canada)
Glutathione	GSH	γ-L-glutamyl-L-cysteinylglycine	C_6_H_11_NO_2_	20 mM	Thermo Fisher
Lysine	Lys	(S)-2,6-Diaminocaproic acid	C_6_H_14_N_2_O_2_	10 mM	Sigma Aldrich
Diaminopimelic acid	DAP	2,6-Diaminopimelic acid	C_7_H_14_N_2_O_4_	30 mM	Sigma Aldrich
Ferulic acid	FA	Trans-ferulic acid	C_10_H_10_O_4_	1 mM ^3^	Sigma Aldrich
Caffeic acid	CFA	(E)-3-(3,4-dihydroxyphenyl) prop-2-enoic acid	C_9_H_8_O_4_	10 mM ^4^	Sigma Aldrich
Benzoic acid	BA	Benzoic acid	C_7_H_6_O_2_	10 mM ^5^	Thermo Fisher
Piperonylic acid	PipA	1,3-benzodioxole-5-carboxylic acid	C_8_H_6_O_4_	3 mM ^6^	Sigma Aldrich

^1^ The chromatogram and mass-spectral information for these metabolites are shown in Appendix A, except for BA and PipA. The identification of BA is tentative, based solely on Tier-3 database annotation. PipA was selected for testing due to its known role as an inhibitor of the phenylpropanoid pathway [69], aiming to confirm the involvement of several DAMs from this pathway that were suppressed during the *Rlm1*-mediated incompatible interaction. ^2^ The maximum concentration without causing visible impact on canola seedlings during a series of pretests. These metabolites were dissolved in deionized water to achieve the designated concentrations unless stated otherwise. ^3,6^ Dissolved initially in DMSO, then diluted with water to achieve 1 mM and 3 mM final concentrations with 0.17% and 1.7% DMSO, respectively. ^4,5^ Dissolved initially in 95% ethanol, then diluted with water to achieve the 10 mM final concentration with 9.5% and 6.8% ethanol, respectively.

## Data Availability

The metabolomics data have been deposited to the MetaboLights [175] with the study identifier MTBLS12542 (https://www.ebi.ac.uk/metabolights/MTBLS12542; deposited on 30 May 2025), following Nature’s Data Repository Guidance.

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
