# Peer review of "Metabolomic Profiling Identifies Key Metabolites and Defense Pathways in Rlm1-Mediated Blackleg Resistance in Canola"

_ijms, 2025, doi:10.3390/ijms26125627_

Round 1
Reviewer 1 Report
Comments and Suggestions for Authors
The manuscript entitled “Metabolomic analysis of Rlm1-based blackleg resistance” is good research in molecular biology and provides good inactive detail in studying Canola's metabolism. However, there are justifications and changes needed.
- I think the title is very short and does not adequately represent the overview of the work done. Please complete the title properly. I suggest adding the plant name used in the experiment.
- In the abstract, what is Rlm1? Please give a short intro so that’s would be easy for the reader to read and understand.
- Why this gene was used against pathogen-resistant needs mentioned in the text? And add the objectives of the study at the end of the introduction section.
- I think there should be no bold results in the results section. All text should not be bold. Please check and confirm.
- Figure 11 quality is very poor. Please revise it. Same for fig.2.
- Writing is fine; add more justification for your results with logic.
Author Response
Thank you for careful evaluation of the manuscript and for helpful suggestions. All minor/editorial suggestions have been adopted. In addition, the Discussion section has been condensed and improved for conciseness and better focus. Brief responses have also been provided to explain how the more substantive comments were addressed, and they are highlighted in blue color.
We believe the revisions have substantially strengthened the work, and we hope the changes are also satisfactory to you.
- I think the title is very short and does not adequately represent the overview of the work done. Please complete the title properly. I suggest adding the plant name used in the experiment.
Response: Agreed and revised.
- In the abstract, what is Rlm1? Please give a short intro so that’s would be easy for the reader to read and understand.
Response: explained.
- Why this gene was used against pathogen-resistant needs mentioned in the text? And add the objectives of the study at the end of the introduction section.
Response: Reasoning provided.
- Figure 1 quality is very poor. Please revise it. Same for fig.2.
Response: Low-resolution graphs were used for initial submission. These figures have been replaced with high-resolution graphs (600 dpi).
- Writing is fine; add more justification for your results with logic.
Response: reasoning/rationale of observations have bee provided for clearer description and interpretation of results and related information in Discussion.
To help guide readers, additional information has been added to subtitles. Section 2.5 has also been reorganized to group DAMs according to their biological pathways and functions into seven focused subsections for better differentiation.
Reviewer 2 Report
Comments and Suggestions for Authors
The manuscript with the title ”Metabolomic analysis of Rlm1-based blackleg resistance”
The introduction presents the importance of the study and mechanism related to pathogen infection in rapeseed.
The aim of the study needs to be clear, please only rephrase it as the study objective or aim, in that way The reader understands the authors have done a report and that’s it, nothing special.
Please add some of you work hypothesis to guide readers throughout the results section.
The materials and methods is very well written and I could find all detail there along with procedure sources. The authors could specify the software for statistics. The results were explored in many interesting ways regarding statistics.
All the results were explored based on a multiple data analysis and the main observations are pointed.
Regarding the discussion sections, the authors have explored theirs results with other studies done in the field.
Please rephrase the conclusion section, remove all citation sources, highlight only the most important conclusions from the study.
Author Response
Thank you for careful evaluation of the manuscript and for helpful suggestions. All minor/editorial suggestions have been adopted. In addition, the Discussion section has been condensed and improved for conciseness and better focus. Brief responses have also been provided to explain how the more substantive comments were addressed, and they are highlighted in blue color.
We believe the revisions have substantially strengthened the work, and we hope the changes are also satisfactory to you.
- The aim of the study needs to be clear, please only rephrase it as the study objective or aim, in that way The reader understands the authors have done a report and that’s it, nothing special.
Response: The aim and objectives have been revised for clearer differentiation.
- Please add some of you work hypothesis to guide readers throughout the results section.
Response: Opening statements have been revised for some of the subsections in Results to better guide readers on hypothesis and purpose of experiment and/or analyses.
- The materials and methods is very well written and I could find all detail there along with procedure sources. The authors could specify the software for statistics. The results were explored in many interesting ways regarding statistics.
Response: The information has been added in section 4.8, Data Analysis. “Most of the data analyses were performed using R and RStudio.”
4: Please rephrase the conclusion section, remove all citation sources, highlight only the most important conclusions from the study.
Response: The conclusion paragraph has been condensed for conciseness and better highlight of key findings.
Reviewer 3 Report
Comments and Suggestions for Authors
Coupled with chemical isotope labeling (CIL) LC-MS and exogenous metabolite application, Zhu et al. identified several plant metabolites and defense-related pathways that are involved in Rlm1-mediated blackleg resistance in canola. This study complements with their prior transcriptomic work. These findings together inform plant breeding strategies aimed at enhancing durable blackleg resistance. The approaches are rigorous, and the results are extensive. However, significant improvements in text clarity are needed before this manuscript can be accepted.
Abstract
- The abstract is informative but overly dense. Consider shortening the list of metabolite classes and instead emphasizing 2–3 key findings.
- Line 13: “PA increased up to 326-fold” – Please specify relative to which condition and confirm that this was statistically significant (e.g., provide adjusted p-values).
Introduction
- The introduction is thorough but slightly repetitive, particularly in the discussion of resistance genes and omics studies. Consider condensing background information to enhance focus on Rlm1-specific knowledge gaps.
- Line 90: “Based on genome-wide transcriptome study…” – This sentence is confusing and needs grammatical correction. Suggest: “A prior transcriptomic study revealed that Rlm1-mediated resistance is non-systemic and activates SA and JA pathways (Zhai et al., 2021).”
Methods
- Line 163–177: The explanation of the isogenic line creation lacks detail on selection criteria for backcrossing generations. Please specify how homozygosity for Rlm1 was confirmed.
- Line 192–194: Indicate how many plants were used per replicate and whether replicates refer to biological or technical replicates.
- Lines 265–275: The mass spectrometry conditions are thorough, but the justification for the chosen gradient and m/z range should be briefly explained (e.g., optimized for what classes of compounds?).
- Line 282: Define “Ratio of Total Useful Signal” at first mention to aid interpretation by readers unfamiliar with CIL LC-MS normalization techniques
- Lines 292–331: The metabolite validation setup is excellent but overly dense. Break into subsections (e.g., “Chemical Preparation”, “Application Procedure”, “Inoculation and Scoring”) for better readability.
- Line 331: You mention experiments were repeated “2–3 times at different intervals.” Please clarify: Were these biological repeats or independent trials? Were all metabolites tested in each repeat?
- Line 344: You use “FDR ≤ 0.05” as a cutoff. Please confirm how multiple hypothesis testing was handled across thousands of metabolites.
- Lines 347–349: The use of ART ANOVA is appropriate, but please state how normality was assessed (e.g., Shapiro-Wilk?).
Results
- Lines 385–395: The reporting of up/downregulated DAMs is very numerical. A summary figure or bar chart would help synthesize these patterns.
- Lines 457–470: The justification for selecting specific DAMs (e.g., PipA, FA, GSH) for functional validation is weak. Are they novel in this context? Previously known to be involved in defense?
- Lines 610–740: Figure 4 and related text are highly detailed. Consider simplifying by grouping DAMs by pathway or function, and move extensive metabolite-level discussion to supplementary.
Discussion
- Line 950: The downregulation of flavonoids is interesting but under-discussed. Are the authors proposing a trade-off mechanism or suppression to avoid pathogen co-option?
Figures and Tables
- Figure 4: Overly dense. Break into separate panels by class (e.g., amino acids, hormones, redox) or by dpi. Add legends with pathway names spelled out.
- Figure 6/7: Add scale bars for every single panels. Include exact concentrations of compounds used in figure legends, not just methods.
Author Response
Thank you for your careful review of the manuscript and for your helpful revision suggestions. All minor and editorial suggestions have been adopted. In addition, the Discussion section has been condensed for greater conciseness and focus. Brief responses have also been provided to explain how the more substantive comments were addressed.
Major changes to the text made in response to substantive comments are highlighted in blue for clarity.
We sincerely appreciate the high-quality review and thoughtful feedback. Your suggestions have been invaluable in refining our analyses and improving the clarity and overall quality of the manuscript. We believe the revisions have substantially strengthened the work, and we hope the changes are satisfactory to you.
- The abstract is informative but overly dense. Consider shortening the list of metabolite classes and instead emphasizing 2–3 key findings.
Response: Abstract has been revised for conciseness and better focus on key findings. The similar approach has also been used to improve the Conclusions.
- The introduction is thorough but slightly repetitive, particularly in the discussion of resistance genes and omics studies. Consider condensing background information to enhance focus on Rlm1-specific knowledge gaps.
Response: Information not closely associated with blackleg impact, Rlm1 or metabolomic analysis has been condensed or eliminated.
- Line 90: “Based on genome-wide transcriptome study…” – This sentence is confusing and needs grammatical correction. Suggest: “A prior transcriptomic study revealed that Rlm1-mediated resistance is non-systemic and activates SA and JA pathways (Zhai et al., 2021).”
Response: Changes made for clarity.
- Line 163–177: The explanation of the isogenic line creation lacks detail on selection criteria for backcrossing generations. Please specify how homozygosity for Rlm1 was confirmed.
Response: Larkan et al. (2016) was cited to avoid duplication, as it describes the use of 43,211 SNP markers to distinguish homozygous (AA or BB), heterozygous (AB), and null (no call) SNP alleles in both Topas DH16516 and the R gene donor parent lines. In our study, we also performed droplet digital PCR (ddPCR) analysis using KASP markers specific to Rlm1, which allowed us to differentiate homozygous and heterozygous individuals. This protocol has not yet been published.
- Line 192–194: Indicate how many plants were used per replicate and whether replicates refer to biological or technical replicates.
Response: This information is in section 4.2; samples from 20 random seedlings of the same treatment were bulked and to form a biological replicate, with three replicates (samples) prepared for each treatment or control at each of the time points.
- Lines 265–275: The mass spectrometry conditions are thorough, but the justification for the chosen gradient and m/z range should be briefly explained (e.g., optimized for what classes of compounds?).
Response: Rationale has been provided.
- Lines 292–331: The metabolite validation setup is excellent but overly dense. Break into subsections (e.g., “Chemical Preparation”, “Application Procedure”, “Inoculation and Scoring”) for better readability.
Response: Suggestion adopted.
- Line 331: You mention experiments were repeated “2–3 times at different intervals.” Please clarify: Were these biological repeats or independent trials? Were all metabolites tested in each repeat?
Response: All metabolites were tested in 2-3 independent trials, with 4 to 24 biological replicates (plants) per treatment and trial, depending on the metabolite and trial. The sentence in 4.7.3 has been modified slightly for clarity.
- Line 344: You use “FDR ≤ 0.05” as a cutoff. Please confirm how multiple hypothesis testing was handled across thousands of metabolites.
Response: The R package DESeq2 was used to perform statistical analysis across the entire metabolite dataset, including correction for multiple hypothesis testing correction using the Benjamini-Hochberg procedure [47]. Slight modification is done to 4.8 for clarity.
- Lines 347–349: The use of ART ANOVA is appropriate, but please state how normality was assessed (e.g., Shapiro-Wilk?).
Response: Yes, the information can be found in 4.8 -Data Analysis.
- Lines 385–395: The reporting of up/downregulated DAMs is very numerical. A summary figure or bar chart would help synthesize these patterns.
Response: The information is summarized with a bar graph (Supplemental Figure S2).
- Lines 457–470: The justification for selecting specific DAMs (e.g., PipA, FA, GSH) for functional validation is weak. Are they novel in this context? Previously known to be involved in defense?
Response: The selection of DAMs was primarily based on their high accumulation in inoculated Rlm1 plants. Previously reported roles in plant resistance were used only as background reference. Some metabolites that accumulated highly in non-inoculated Rlm1 plants—such as ferulic acid, caffeic acid and benzoic acid—were also tested. However, the roles for almost all these selected metabolites have not been previously described or validated for blackleg resistance in canola. The section “4.7.1 Chemical (metabolite) preparation” has been revised to clarify the rationale for metabolite selection.
- Lines 610–740: Figure 4 and related text are highly detailed. Consider simplifying by grouping DAMs by pathway or function, and move extensive metabolite-level discussion to supplementary.
Response: Figure 4 presents DAMs grouped generally by the pathway or functional category. Given the complexity of the data and the importance of showing clear biological trends at different time points, we exported the heatmaps directly from the metabolomics analysis software, which, unfortunately, limited our ability to further modify the visual effects.
However, we’ve restructured the heavy text into seven biologically coherent subsections (Sections 2.5.1 to 2.5.7) to enable a more streamlined and thematic presentation of DAMs. Where relevant, the description emphasizes pathway-level interpretation over individual metabolite changes to reduce complexity in the main text while preserving transparency and reproducibility.
- Line 950: The downregulation of flavonoids is interesting but under-discussed. Are the authors proposing a trade-off mechanism or suppression to avoid pathogen co-option?
Response: We appreciate this insightful comment. Yes, we proposed a potential trade-off mechanism whereby the downregulation of certain flavonoids may help limit exploitation by the pathogen. This point is in the final paragraph of the Discussion section. Although direct evidence is limited, there is growing recognition that some specialized metabolites can be co-opted by pathogens. Therefore, the suppression of flavonoid biosynthesis in the resistant interaction (RI) may represent a strategic reallocation of resources or a defense optimization strategy. We acknowledge this as a plausible explanation and an important avenue for future research.
- Figure 4: Overly dense. Break into separate panels by class (e.g., amino acids, hormones, redox) or by dpi. Add legends with pathway names spelled out.
Response: As noted in our response to Comment 15, we have limited flexibility to modify the heatmaps generated directly from the metabolomics analysis software. However, to improve interpretability, we have restructured the relevant text into seven biologically meaningful subsections. We hope this compensates for the visual constraints and helps guide the reader through the key findings more effectively.
16: Figure 6/7: Add scale bars for every single panels. Include exact concentrations of compounds used in figure legends, not just methods.
Response: Figures 6 and 7 are composed of original, unscaled camera images, with each panel representing a single experimental setup. Control and treatment cotyledons were photographed side by side under identical conditions to highlight relative lesion sizes. While scale bars can enhance precision, we believe they are less critical in this context, as the images are intended for qualitative comparison. Additionally, accurate scale bars are not possible post hoc without the original samples.
We have now added compound concentrations in the figure legends, as suggested. This information is also available in the Materials and Methods section (Table 4).
Reviewer 4 Report
Comments and Suggestions for Authors
Manuscript ID:3595597
Title: Metabolomic analysis of Rlm1-based blackleg resistance
The study explores the biochemical mechanisms of Rlm1-mediated blackleg resistance by comparing a susceptible double haploid (DH) line of Topas with its resistant isogenic counterpart. The authors identified key primary metabolites, defense-related pathways, and post-transcriptional mechanisms using chemical isotope labeling liquid chromatography-mass spectrometry.
The introduction can be improved by providing information on canola production, the impact of blackleg on canola yield, and the role of amino acids/pipecolic acid, secondary metabolites in disease resistance. Please clearly describe the objectives (i.e., 3, 7, and 11 dpi). The methods are adequately described, and the results are well discussed.
Supplementary File
Figure S1-3 legends are missing.
Please provide LC-MS data for the detected compounds as supplementary information.
Other comments:
Please italicize the species name, e.g., Botrytis cinerea.
Abstract: Should be self-explanatory; include the methods.
‘DH’ line: Please expand the abbreviation. ‘Double haploid’ (DH)
Introduction
‘some secondary metabolites play key roles in plant resistance against pathogens [22,23]’ Please indicate the secondary metabolites.
Materials and methods
‘about 65% relative humidity and a daily photoperiod of 16 h.’ Please indicate the light intensity.
‘at 20°C under cool-white fluorescent light for 7-10 d’ Please indicate the light intensity.
Results
Please identify linear relationships within the metabolic dataset among the samples.
Please provide the LC-MS chromatograms and spectra for the metabolites present in Table 1.
Author Response
Thank you for your careful review of the manuscript and for your helpful revision suggestions. All minor and editorial suggestions have been adopted. In addition, the Discussion section has been condensed for greater conciseness and focus. Brief responses have also been provided to explain how the more substantive comments were addressed.
Major changes to the text made in response to substantive comments are highlighted in blue for clarity.
We sincerely appreciate the high-quality review and thoughtful feedback. Your suggestions have been invaluable in refining our analyses and improving the clarity and overall quality of the manuscript. We believe the revisions have substantially strengthened the work, and we hope the changes are satisfactory to you.
- The introduction can be improved by providing information on canola production, the impact of blackleg on canola yield, and the role of amino acids/pipecolic acid, secondary metabolites in disease resistance. Please clearly describe the objectives (i.e., 3, 7, and 11 dpi). The methods are adequately described, and the results are well discussed.
Response: The information has been provided in the introduction as suggested.
- Please provide LC-MS data for the detected compounds as supplementary information.
Response: Supplemental Table S1 has been added for the information.
- Abstract: Should be self-explanatory; include the methods.
Response: The abstract has been revised to strike the right balance of information, including the addition of key analytical methodologies for the metabolomic study
- ‘some secondary metabolites play key roles in plant resistance against pathogens [22,23]’ Please indicate the secondary metabolites.
Response: PA, terpenes and GLS have been cited and discussed as examples.
- Please identify linear relationships within the metabolic dataset among the samples.
Response: Thank you for this comment. While our analysis did not explicitly focus on identifying linear correlations between individual metabolites across samples, we used Principal Component Analysis (PCA) to reveal overall variance structure and clustering patterns among sample groups (Figure S1). PCA is based on linear combinations of variables and thus captures major linear trends in the dataset. If needed, we can explore pairwise correlation matrices or other linear modeling approaches in future work to investigate metabolite-level relationships more specifically.
- Provide the LC-MS chromatograms and spectra for the metabolites present in Table 1.
Response: Supplemental Figure S4-11 have been added for the information.
Reviewer 5 Report
Comments and Suggestions for Authors
Dear authors,
I carefully reviewed your manuscript and found that the following points need to be improved:
1. Why Rlm1 gene was considered in this study? The best was that authors may check a list of candidate genes and then choose the best. However, please justify why this gene was chosen?
2. Basically, the abstract should begin with problem statement. Please write the problem statement at the beginning of the introduction.
3. Please rearrange the keyword based on alphabetical order.
4. The introduction should have a logical flow, explaining the importance of Canola, then blackleg and…..
5. In the introduction please explain how severe its blackleg in Canada.
6. I think the last paragraph of the introduction is just irrelevant and can be removed.
7. Please provide a concrete hypothesis for this study at the end of introduction.
8. Please strengthen the discussion section by comparing your findings with more previous studies and please provide the similarities and differences. The similarities will strengthen your findings and the differences will indicate the significance and novelty of your findings.
9. In materials and methods please provide the details of the devices or machineries used in this research. Informations such as manufacturer, city and country are required.
10. In statistical analysis please provide the version and other details R or Rstudio software used for analyzing the data.
11. In results section 3.1, please provide the PCA plot right in this section rather than in supplementary.
12. Why the authors performed both multivariate and univariate analysis?
13. What is the difference between figure 1 and figure 2 I suggest that including them in one figure would be better.
14. From tables 2-4 please keep one which best fits and the others move them to supplementary data.
15. Figure 5, why 3 days after inoculation was considered rather than starting from 1 day after inoculation?
16. Figure 6, why 3 picture of cotyledons in each treatment? It’s confusing because each shows a different symptom. If each is for a different purpose then it should be indicated on the figure. Same applies on figure 7.
17. I suggest to write the discussion in a better way. Please do a compare and contrast of your findings with previous studies. While the similarities confirms the findings, the differences will indicate the significant of your findings.
18. Please add two more sections in this manuscript. After discussion please add two sections one should explain the limitations and other future perspectives.
19. In the conclusion please indicate whether the objectives of the studies were achieved and the answer for the hypothesis.
20. Please always use a uniform format of citation and referencing.
21. If the authors can make a visual abstract it would be much interesting.
22. I also suggest not overload the manuscript with big number of unnecessary references.
Author Response
Thank you for your careful review of the manuscript and for your helpful revision suggestions. All minor and editorial suggestions have been adopted. In addition, the Discussion section has been condensed for greater conciseness and focus. Brief responses have also been provided to explain how the more substantive comments were addressed.
Major changes to the text made in response to substantive comments are highlighted in blue for clarity.
We sincerely appreciate the high-quality review and thoughtful feedback. Your suggestions have been invaluable in refining our analyses and improving the clarity and overall quality of the manuscript. We believe the revisions have substantially strengthened the work, and we hope the changes are satisfactory to you.
1: Why Rlm1 gene was considered in this study? The best was that authors may check a list of candidate genes and then choose the best. However, please justify why this gene was chosen?
Response: We chose to focus on Rlm1 because it is one of the most widely deployed blackleg resistance genes in canola breeding programs globally, making it highly relevant for both scientific understanding and practical application. This study also builds on our earlier transcriptomic analysis of Rlm1-mediated resistance (Zhai et al., 2021), allowing us to now explore the associated defense-related metabolites and pathways in more detail. We have provided this rationale more explicitly in the revised Introduction section. While we have also completed metabolomic studies on other blackleg resistance genes—including those involved in quantitative resistance—this manuscript presents our first set of findings, with additional results to be reported separately.
- Basically, the abstract should begin with problem statement. Please write the problem statement at the beginning of the introduction.
Response: Agreed and revised.
- The introduction should have a logical flow, explaining the importance of Canola, then blackleg and…..
Response: Revised for better flow and focus.
- I think the last paragraph of the introduction is just irrelevant and can be removed.
Response: The paragraph in question was intended to briefly introduce chemical isotope labeling (CIL) LC-MS, emphasizing its advantages over conventional metabolomics approaches—particularly its ability to enhance metabolite quantification and detection sensitivity. This context helps set the stage for the more detailed explanation provided in the Materials and Methods section. To alleviate the concern and also avoid redundancy, we have pared down this information in the Introduction. In addition, we have added two new paragraphs: one summarizing relevant plant defense-related metabolomics findings, and another more clearly outlining the aims and objectives of the study.
- Please provide a concrete hypothesis for this study at the end of introduction.
Response: Addressed. Additionally, the aims and objectives are now more clearly outlined at the end of Introduction.
- Please strengthen the discussion section by comparing your findings with more previous studies and please provide the similarities and differences. The similarities will strengthen your findings and the differences will indicate the significance and novelty of your findings.
Response: We appreciate the suggestion. We have carefully revised the Discussion section to include more comparisons between our findings and those of previous studies. Where possible, we highlight similarities that support our results, as well as differences that underscore potential novelty of our findings. For several metabolites, however, there are limited or no prior reports linking them to plant disease resistance, which we have also acknowledged in the revised text.
- In results section 3.1, please provide the PCA plot right in this section rather than in supplementary.
Response: We considered including the PCA plot in the main text, but given the number of existing figures and tables, we felt that placing it in the supplementary materials helps maintain a balanced and focused presentation in the main manuscript. However, we are open to move the PCA plot into the main Results section if the editors or reviewers feel it would improve clarity and accessibility.
- Why the authors performed both multivariate and univariate analysis?
Response: We performed both analyses to obtain complementary insights; multivariate methods (e.g., PCA, PLS-DA) were used to explore overall data structure, identify patterns, and assess group separation across samples, while univariate methods (e.g., t-tests, fold change analysis) were applied to pinpoint individual metabolites significantly contributing to these differences. This integrated approach allows us to gain both global (pattern-level) and specific (feature-level) understanding of the data. Rationale has been provided for using both multivariate and univariate analysis in the Material and Method section.
- What is the difference between figure 1 and figure 2? I suggest that including them in one figure would be better.
Response: Figure 1 illustrates the up- and down-regulated DAMs across different time points within the same treatment, emphasizing temporal dynamics. In contrast, Figure 2 compares DAMs between different treatments at the same time point, highlighting treatment-specific effects. The clarification has been added in figure captions to avoid confusion. Given these distinct comparative frameworks, presenting them as separate figures likely helps maintain clarity.
- From tables 2-4 please keep one which best fits and the others move them to supplementary data.
Response: These tables highlight key pathways identified at different time points, helping to illustrate how pathway involvement evolves in relation to plant resistance. We believe retaining them in the main text supports the temporal interpretation of the results. To aid clarity, we have marked the corresponding time points in bold within each table title. We are also open to moving some of them to Supplementary Data if preferred by the editors.
- Figure 5, why 3 days after inoculation was considered rather than starting from 1 day after inoculation?
Response: 3 dpi was selected because it represents a critical stage for L. maculans to establish a transient biotrophic relationship with the host during the infection process, before transitioning to the necrotrophic phase around 7–10 dpi, when visible necrotic lesions begin to form (Li et al. 2004; Zhai et al. 2021). This rationale and relevant references have been added following the description of post-inoculation sampling intervals in the Materials and Methods section (page 20).
- Figure 6, why 3 picture of cotyledons in each treatment? It’s confusing because each shows a different symptom. If each is for a different purpose then it should be indicated on the figure. Same applies on figure 7.
Response: Three inoculated cotyledons were shown for each treatment to illustrate the range of symptoms observed under the same condition (reflecting biological variability). A clarification has been added to the figure captions to indicate this purpose.
- I suggest to write the discussion in a better way. Please do a compare and contrast of your findings with previous studies. While the similarities confirms the findings, the differences will indicate the significant of your findings.
Response: We appreciate the suggestion. We have carefully revised the Discussion to include more comparisons between our findings and those of previous studies. Where possible, we tried to highlight similarities, as well as differences. Some metabolites have limited or no prior reports linking them to plant disease resistance, which we have also acknowledged in the revised text.
- Please add two more sections in this manuscript. After discussion please add two sections one should explain the limitations and other future perspectives.
Response: Two additional sections have been added in response to the suggestion.
- In the conclusion please indicate whether the objectives of the studies were achieved and the answer for the hypothesis.
Response: Suggestion adopted and the information has been added.
- If the authors can make a visual abstract it would be much interesting.
Response: Agreed, and will include a graphical abstract with the final submission..
- I also suggest not overload the manuscript with big number of unnecessary references.
Response: We have gone through the manuscript and tried to retain only essential references.
Round 2
Reviewer 3 Report
Comments and Suggestions for Authors
This is the revised manuscript by Zhu et al. In this version, the authors have effectively addressed my comments. I am satisfied with their responses and hence recommend it for acceptance.
Reviewer 5 Report
Comments and Suggestions for Authors
I carefully evaluated the revised version of the manuscript. Although there are still few gaps however the authors addressed the comments and made the reasonable justifications that satisfied me.